# Integration of partially observed multimodal and multiscale neural signals for estimating a neural circuit using dynamic causal modeling

**Jiyoung Kang**[1,2], **Hae-Jeong Park**[2,3,4,5]*

1 Department of Scientific Computing, Pukyong National University, Busan, Republic of Korea, 2 Center for Systems and Translational Brain Sciences, Institute of Human Complexity and Systems Science, Yonsei University, Seoul, Republic of Korea, 3 Graduate School of Medical Science, Brain Korea 21 Project, Department of Nuclear Medicine, Psychiatry, Yonsei University College of Medicine, Seoul, Republic of Korea, 4 Department of Cognitive Science, Yonsei University, Seoul, Republic of Korea, 5 Brain Research Institute, Institute for Innovation in Digital Healthcare, Yonsei University College of Medicine, Seoul, Republic of Korea

* parkhj@yonsei.ac.kr

## Abstract

Integrating multiscale, multimodal neuroimaging data is essential for a comprehensive understanding of neural circuits. However, this is challenging due to the inherent trade-offs between spatial coverage and resolution in each modality, necessitating a computational strategy that combines modality-specific information effectively. This study introduces a dynamic causal modeling (DCM) framework designed to address the challenge of combining partially observed, multiscale signals across a larger-scale neural circuit by employing a shared neural state model with modality-specific observation models. The proposed method achieves robust circuit inference by iteratively integrating parameter estimates from local microscale and global meso- or macroscale circuits, derived from signals across various scales and modalities. Parameters estimated from high-resolution data within specific regions inform global circuit estimation by constraining neural properties in unobserved regions, while large-scale circuit data help elucidate detailed local circuitry. Using a virtual ground truth system, we validated the method across diverse experimental settings, combining calcium imaging (CaI), voltage-sensitive dye imaging (VSDI), and blood-oxygen-level-dependent (BOLD) signals—each with distinct coverage and resolution. Our reciprocal and iterative parameter estimation approach markedly improves the accuracy of neural property and connectivity estimates compared to traditional one-step estimation methods. This iterative integration of local and global parameters presents a reliable approach to inferring extensive, complex neural circuits from partially observed, multimodal, and multiscale data, showcasing how information from different scales reciprocally enhances entire circuit parameter estimation.

**Data Availability Statement:** Codes used in the current study are available at the github: https://github.com/monet-yonsei/mmsdcm.

**Funding:** This research was supported by Brain Research Program through the National Research Foundation of Korea (NRF) funded by the Ministry of Science and ICT (Grant No. 2023R1A2C2006217 to HJP), the Bio & Medical Technology Development Program of the NRF (Grant No. RS-2024-00401794 to HJP), and the Basic Science Research Program of the NRF funded by the Ministry of Education of Korea (Grant No. 2021R1I1A1A01059755 to JK). The funders had no role in study design, data collection and analysis, decision to publish, or preparation of the manuscript.

**Competing interests:** The authors have declared that no competing interests exist.

## Author summary

Reliable estimation of a computational neural circuit model requires integrating data from various brain imaging techniques, each providing unique but limited insights into brain activity at different spatial and temporal scales. Combining these multiscale, multimodal data sources is challenging due to trade-offs between spatial coverage and resolution inherent in each modality. In this study, we introduce a novel dynamic causal modeling (DCM) framework that overcomes these challenges by allowing partially observed data across scales to jointly inform the estimation of neural circuit parameters. Our method is distinctive in its use of a shared neural state model, paired with modality-specific observation models, to iteratively integrate local, high-resolution data and global, lower-resolution data. This reciprocal approach leverages detailed local circuit information to constrain parameter estimation for unobserved regions within the broader network, while also using global circuit data to refine local circuit estimates. This iterative, multiscale integration enables more accurate circuit inference than traditional one-step methods, which typically struggle with sparse data and complex neural structures. By demonstrating how information from different scales and modalities can complement each other, our framework provides a powerful tool for reconstructing neural circuits from incomplete data. This approach has broad implications for advancing our understanding of complex neural systems, particularly in preclinical research, where comprehensive, multiscale neural data are often scarce.

## 1 Introduction

The brain operates as a multiscale system characterized by sophisticated interactions at both local and global levels. Locally, interactions among regionally confined neural cells form functional units within the broader context of global interactions. Global interactions, in turn, provide neural contexts that influence the dynamics within local circuits [1].

Direct measurement of interactions at any scale is not feasible; instead, we need to infer these interactions from activities observed in the neural system. To infer asymmetric bidirectional interactions, referred to as effective connectivity, it is essential to use a computational model with a biologically grounded connectivity topology. The challenge then becomes reliably estimating the connectivity parameters of the model using observed signals. This is where the diversity of neuroimaging modalities plays a crucial role. Each modality provides a unique perspective on neural activity, contributing complementary data that enrich the process of model parameter estimation, leading to more plausible solutions to accessing the underlying neural circuit. However, integrating imaging data across different scales and modalities to enhance model parameter estimation is not trivial. This is particularly critical given the inherent limitations of conventional imaging techniques, which often involve trade-offs between spatial and temporal resolution and scope. Multimodal signals with different scales fail to capture data concurrently over the same spatial extent, leading to partial data and lacking signals in some areas. Thus, integrating such multimodal signals necessitates a novel technique that can effectively combine distinct but partial information with a different data representation (see Fig 1A).

Recent technological advancements have improved the acquisition of multimodal multiscale neuroimaging data, leveraging the specific strengths of each modality, particularly in preclinical research. In the preclinical animal study, neuroimaging methods like calcium imaging (CaI), voltage-sensitive dye imaging (VSDI), and functional magnetic resonance imaging

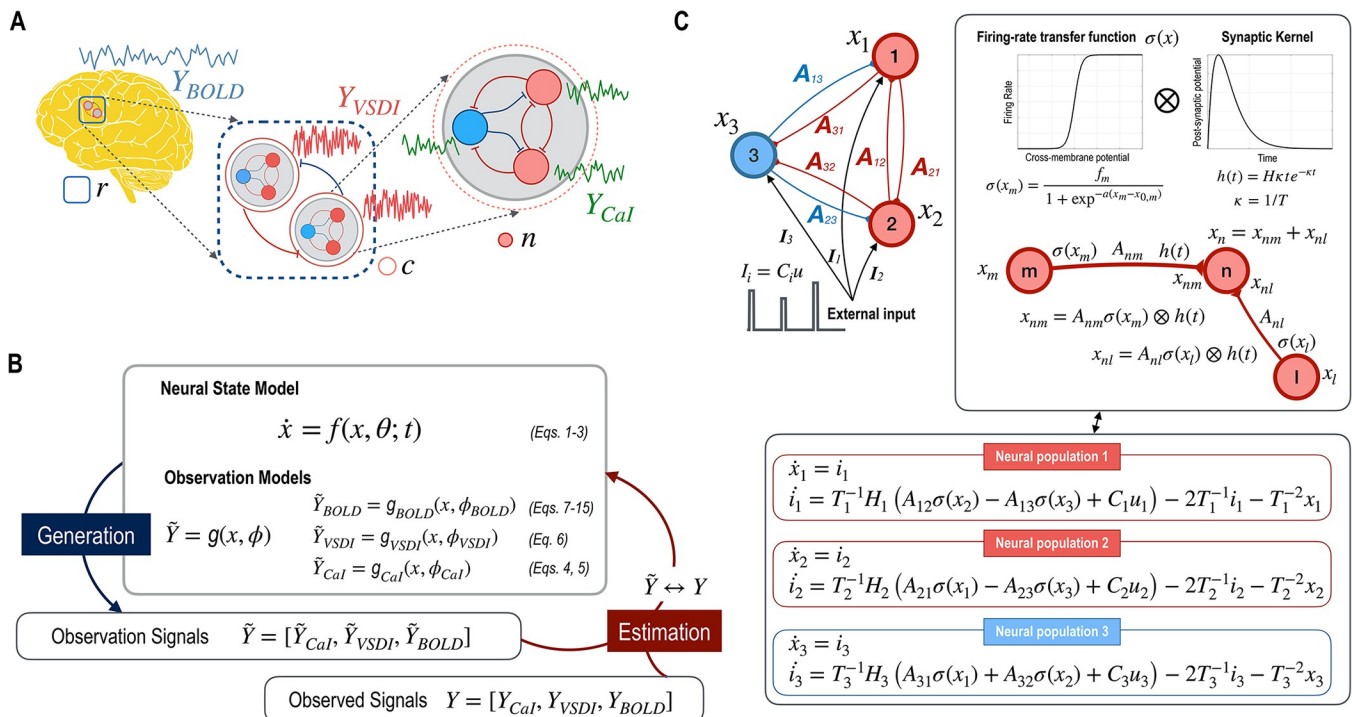

**Fig 1. Multimodal and multiscale signals and parameter estimation with mms-DCM.** A. Three neuroimaging modalities of CaI, VSDI, and BOLD ($Y_{CaI}$, $Y_{VSDI}$, and $Y_{BOLD}$), have different temporal and spatial resolutions in the current experimental setting. CaI measures activities within a neural population ($n$) with a middle temporal resolution (0.1 sec), VSDI measures activities among cortical columns ($c$) with high temporal resolution (1 msec), and BOLD measures activities of a region ($r$) composed of multitudes of neural columns with low temporal resolution (1 sec). B. In mms-DCM, a common neural state dynamics denoted as $f$ generates CaI, VSDI, and BOLD ($Y_{CaI}$, $Y_{VSDI}$, and $Y_{BOLD}$) through corresponding observation model functions ($g_{CaI}$, $g_{VSDI}$, and $g_{BOLD}$). By fitting multimodal observed signals, $Y = [Y_{CaI}, Y_{VSDI}, Y_{BOLD}]$ with model generated signals $\tilde{Y} = [\tilde{Y}_{CaI}, \tilde{Y}_{VSDI}, \tilde{Y}_{BOLD}]$ mms-DCM estimates model parameters for the common neural state model $f$ and each observation model function ($g_{CaI}$, $g_{VSDI}$, and $g_{BOLD}$). C. A schematic of convolution-based neural state model for two excitatory neural populations (1, 2) and one inhibitory neural population (3) is presented. For every connection from a source ($m$) to target ($n$) neural populations, an effective connectivity ($A_{nm}$)−weighted firing-rate transfer function of a membrane potential $\sigma(x_m)$ of a source neural population is convoluted with the synaptic kernel ($h(t)$) to affect the membrane potential of the target neural population ($x_{nm}$).

(fMRI) are increasingly used. CaI, measuring photon emissions reflecting intracellular $Ca^{2+}$ ion concentrations, reveals the activation of single or multiple neurons. VSDI offers insights into membrane potentials with precise spatial and temporal resolution. fMRI detects blood-oxygen-level-dependent (BOLD) signals across the whole brain, providing broader, albeit less detailed, neural activity patterns. Researchers have begun to combine two or more of these signal types synchronously, taking advantage of their respective strengths. For instance, Schlegel proposed a fiber-optic implant-based method for simultaneously detecting CaI and BOLD fMRI signals in mice [2]. Cecchetto integrated CaI with multichannel local field potential (LFP) recordings, addressing the limitations of CaI in capturing subthreshold neural activities [3].

To infer the effective connectivity of a neural circuit model using multimodal and multiscale data in preclinical research, we employed the dynamic causal modeling (DCM) framework [4,5]. This approach integrates a distinct neural state (a connectivity-based circuit) and an observation model. Traditionally, DCM has been used to model neural circuits from single modality signals like LFP, electroencephalograms (EEG), and fMRI [6–9]. More recently, DCM has been adapted to incorporate multimodal data for brain circuit analysis. For instance, Wei et al. [10] introduced a multimodal DCM that combines the high temporal resolution of EEG with the high spatial resolution of fMRI data in human research. Their approach utilizes

separate neural state models for EEG and fMRI and leverages EEG-derived model parameters to estimate macroscopic neural interactions via fMRI [10]. Many preclinical applications, however, require a method to estimate parameters of a single neural state model by combining partial data across modalities with vastly different scales.

In the current study, we introduce a multimodal multiscale DCM (mms-DCM) framework to address the challenge of combining partially observed signals to estimate an extensive neural circuit. The mms-DCM encompasses a single neural state and multiple observation models to accommodate heterogeneous signal types. We hypothesized that information from the large-scale circuit, with broader spatial coverage, can be instrumental in deciphering local circuitry. Conversely, we also hypothesized that the detailed properties of a local circuit, captured in high-resolution imaging, are essential for modeling large-scale interactions by refining nodal properties within the larger network. To achieve precise inference of neural circuitry with wide coverage, we employed reciprocal integration of local and global information from different signal scales and modalities. The parameters of local circuits, inferred from high-resolution signals in specific regions, were used as priors for unobserved regions to estimate larger circuit parameters from lower-resolution imaging and vice versa in an iterative procedure.

We evaluated the mms-DCM framework in diverse virtual experiments, constructed using simulated data from a computational model of the L2/3 mouse barrel cortex, derived from experimental CaI data [1]. These virtual experiments involved various combinations of simulated observation signals, including CaI, VSDI, and fMRI, to reflect conditions relevant to a preclinical research environment.

In the next three sections, we introduce the mms-DCM framework (section 2), construct a ground truth system for virtual experiments based on experimental data (section 3), and demonstrate its model parameter estimation performance across different combinations of signal modalities in virtual experimental settings (section 4).

By illustrating how signals at the micro- and mesoscopic scales reciprocally inform each level of neural circuit estimation, we demonstrate the framework's capability to provide plausible solutions to real-world experimental problems.

## 2 Method I: Introduction of mms-DCM

### 2.1. Virtual experimental settings

In the mms-DCM framework, we focus on three key imaging modalities commonly or increasingly used in animal studies: CaI, VSDI, and BOLD signals, respectively labeled as $Y_{CaI}$, $Y_{VSDI}$, and $Y_{BOLD}$. Each of these modalities offers distinct spatial and temporal resolutions. For instance, advanced CaI technology can achieve near-millisecond temporal resolution, but typical CaI signals are captured at about 10–15 Hz, or approximately 0.1 seconds [11]. VSDI provides optical imaging of membrane potentials with a spatial resolution of 20–50 μm and a temporal resolution of 1–2 ms [12–14]. In contrast, fMRI detects BOLD signals over the entire brain but with slower temporal resolution (1–2 seconds) and coarser spatial resolution (about millimeters).

In our virtual experimental setup, we consider CaI for high spatial resolution but limited scale coverage (focusing on a set of neurons, $n$), VSDI for medium spatial resolution over a moderate area (covering neural populations in a column, $c$), and BOLD fMRI for broad-scale, low spatial resolution imaging (encompassing neural populations in a region, $r$). We also account for the varying temporal resolutions of these modalities: approximately 1 ms for VSDI, 0.1 second for CaI, and 1 second for BOLD fMRI. This is explained in Fig 1A.

The mms-DCM framework extends the conventional DCM scheme, which is implemented in the SPM12 toolbox (https://www.fil.ion.ucl.ac.uk/spm/) [4,5]. As shown in Fig 1B, the

mms-DCM encompasses a shared neural state dynamics model $f$ and three observation models ($g_{CaI}, g_{VSDI}$, and $g_{BOLD}$) to simulate CaI, VSDI, and BOLD ($Y_{CaI}$, $Y_{VSDI}$, and $Y_{BOLD}$), the details of which will be explained in the following sections.

## 2.2. Neural state dynamics model: a convolution-based neural population model

In our study, we employed a convolution-based neural population model [8,15] to describe neural state dynamics. This model has widely been used to model neural circuits with electrophysiological data such as LFP or EEG/MEG [6,8] and has been extended for CaI [1], VSDI [16] and fMRI data [10,17,18]. Here, we briefly introduce the convolution-based neural state model. For a detailed explanation, please refer to the review by [8].

In the convolution-based model, the activity of the presynaptic neural population is transformed into a firing rate through a sigmoidal function. Specifically, the membrane potential $x_m$ at the neural population $m$ affects the membrane potential at the neural population $n$ by the convolution of the firing-rate transfer function in the form of sigmoidal function $\sigma(x_m)$ and the synaptic kernel $h(t)$ (Fig 1C),

$$x_n = \sum_m A_{nm}\sigma(x_m) \otimes h(t),$$

where the effective connectivity from a neural population $m$ to a neural population $n$ is denoted by $A_{nm}$. The sigmoidal (activation) function $\sigma(x_m)$ of a neural population $m$ transforms the membrane potential $x_m$ to the firing rate of action potentials, denoted by:

$$\sigma(x_m) = \frac{f_m}{1 + exp^{-a(x_m - x_{0,m})}} \tag{1}$$

, where $f_m$ and $a$ represent a maximal firing rate and a slope of the sigmoid function. Parameter $x_{0,m}$ is the postsynaptic potential (PSP) that achieves a 50% firing rate of a neural population $m$ (Jansen and Rit, 1995). The synaptic kernel $h(t)$ is described as,

$$h(t) = H_n\kappa_n te^{-\kappa_n t}, \text{ where } \kappa_n = 1/T_n.$$

Here, $T_n$ and $H_n$ are the decay time constant and the maximal PSP of a neural population $n$. Mathematically, this relationship can be rewritten as the following ordinary differential equations of the cross-membrane current $i_n$ of a neural population $n$:

$$\frac{dx_n}{dt} = i_n, \tag{2}$$

$$\frac{di_n}{dt} = T_n^{-1}H_n\left(\sum_{m=1}^{N} A_{nm}s_m\sigma(x_m) + C_n u_n\right) - 2T_n^{-1}i_n - T_n^{-2}x_n, \tag{3}$$

The polarity of a neural population $n$ is indicated by $s_n$, which is +1 for excitatory neural populations and -1 for inhibitory neural populations. The external input to neural populations $u_n(t)$ is multiplied by the input modulation parameter $C_n$. In neural population $n$, the rate of change in cross-membrane current is proportional to the sum of the weighted external input $u_n$ and all incoming neurons' firing rates weighted by the effective connectivity $A_{nm}$ and its polarity $s_m$.

All parameters are summarized in Table 1.

**Table 1. Model parameters in the neural state model.**

| Parameter | Value* | |
|---|---|---|
| Resting membrane potential $V_{rest}$ | -65 mV | |
| Decay time constant $T_n$ | $T_0 = 0.128$ s <br> $T_0\exp(\theta_i)$ | $\theta_i \sim N(0, 1/256)$ |
| Threshold for action potential $V_{th}$ | -40 mV | |
| Maximal postsynaptic potential $H$ | 27.18 mV | |
| Maximal firing rate $f$ | 30 Hz | |
| Effective connectivity** $A_{mn}$ | $A_0 = 0.17$ <br> $A_0\exp(\theta_i)$ | $\theta_i \sim N(0, 1/32)$ |
| Input modulation parameter** $C_n$ | $C_0 = 0.25$ <br> $C_0\exp(\theta_i)$ | $\theta_i \sim N(0, 1/32)$ |

*The Gaussian priors distributions are presented as $N$(mean, variance).

** Reference values, $A_0$ and $C_0$, were determined by parameter optimization in constructing the ground truth system described in section 3.

## 2.3. Combinations of observation models

Observation models transform neural population activity into the observed signals ($Y_{\text{CaI}}$, $Y_{\text{VSDI}}$, and $Y_{\text{BOLD}}$). The subsequent subsections explain details of the observation models for CaI, VSDI, and BOLD fMRI signals.

**2.3.1. Observation model for CaI signals.** Following previous studies with CaI signals [1,19], the state dynamics equation for calcium concentration $[Ca^{2+}]_n$ of a neuron $n$ [20],

$$\frac{d}{dt}[Ca^{2+}]_n = -k_{\text{Ca}}g_{\text{Ca}} \cdot (x_n - E_{\text{Ca}}) \cdot \sigma(x_n - V_{\text{HVA}}) - \frac{[Ca^{2+}]_n - [Ca^{2+}]_{\text{base}}}{\tau_{\text{Ca}}}, \tag{4}$$

is used. Here, the parameter $k_{\text{Ca}}$ is the conversion ratio from a calcium ion current to its concentration per time unit; $g_{\text{Ca}}$ represents the maximal conductance of calcium ions, $E_{\text{Ca}}$ is the reversal potential of calcium ion, the membrane potential $x_n$ in Eqs 2 and 3 is an input to the calcium ion concentration dynamics, and $\tau_{Ca}$ is a time constant for calcium decay. Following Rahmati et al. [19], we employed a high-voltage-activated calcium channel (HVA) $V_{\text{HVA}}$ in the sigmoidal function, $\sigma(x - V_{\text{HVA}})$.

We transformed the calcium ion concentration $[Ca^{2+}]_n$ into the CaI signal $Y_{CaI,n}$ with the saturating Hill-type function [1,19]:

$$Y_{CaI,n} = g_{CaI}(x, \phi_{CaI}) = k_F \frac{[Ca^{2+}]_n}{[Ca^{2+}]_n + K_d} + d_F, \tag{5}$$

where $k_F, K_d$, and $d_F$ represent scale, dissociation, and offset parameters. The parameters in Eqs 5 and 6 that describe calcium dynamics were obtained from previous studies and are described in Table 2 [1,19,21].

**2.3.2. Observation model for VSDI signals.** We use a linear model between membrane potential and VSDI signals [16], as reported in previous studies [22,23]. The VSDI signals are given by a linear weighted sum of the membrane potential of neural populations:

$$Y_{\text{VSDI},c} = g_{VSDI}(x, \phi_{VSDI}) = \alpha \sum_{n \in [N_c]} \rho_n x_n. \tag{6}$$

Here, $Y_{\text{vsdi},c}$ represents the VSDI signal at a cortical column $c$, composed of $[N_C]$ neural populations and is a weighted sum of the membrane potential $x_n$ at neural population $n$ with the contribution ratio $\rho_n$ to the VSDI signal in column $c$. The $\alpha$ is a scaling parameter for the

**Table 2. Parameters in observation models.**

| | | Parameter | Value |
|---|---|---|---|
| $\phi_{CaI}$ | $k_{Ca}$ | Conversion ratio from a calcium ion current to its concentration per time unit* | 0.18 |
| | $g_{Ca}$ | Maximal conductance of calcium ions | 5 mS/cm |
| | $E_{Ca}$ | Reversal potential of calcium ion | 120 mV |
| | $V_{HVA}$ | Membrane potential that achieves 50% of high-voltage-activated calcium channel (HVA) in the sigmoidal function, $\sigma(v_n - V_{HVA})$ | -27.89 mV |
| | $\tau_{Ca}$ | Time constant for calcium decay* | 1.44 sec |
| | $[Ca^{2+}]_{base}$ | Calcium concentration at rest | 100 nM |
| | $k_F$ | Scale parameter | 9.85 |
| | $K_d$ | Dissociation parameter | 200 nM |
| | $d_F$ | Offset parameter | -3.283 |
| $\phi_{VSDI}$ | $\alpha$ | Scaling factor for VSDI signals | 0.01 |
| | $\rho_n$ | Contribution ratio $\rho_n$ to the VSDI signal in column $c$ of the membrane potential at neural population $n$ | 0.8 for excitatory 0.2 for inhibitory |
| $\phi_{BOLD}$ | $\beta_{n,exc}$ | Scaling factor for excitatory input in neurovascular coupling | 0.1 |
| | $\beta_{n,inh}$ | Scaling factor for inhibitory input in neurovascular coupling | 0.1 |
| | $\beta_{n,ext}$ | Scaling factor for external input in neurovascular coupling | 0.1 |
| | H | Rate of vasodilatory signal decay per sec | 0.64 |
| | $\chi$ | Rate of flow-dependent elimination | 0.32 |
| | $\tau$ | Rate hemodynamic transit per sec | 2.0 |
| | $\alpha$ | Grubb's exponent | 0.32 |
| | $\varphi$ | Resting oxygen extraction fraction | 0.40 |
| | $V_0$ | Blood volume fraction | 4 |
| | $k_1$ | Intravascular coefficient | 2.773 |
| | $k_2$ | Concentration coefficient | 1.087 |
| | $k_3$ | Extravascular coefficient | -1.718 |

* In the estimation for the construction of the virtual model, we optimized $k_{Ca}$ and $\tau_{Ca}$ in the parameter optimization step with DCM, and the final estimated values used in the virtual model are shown.

VSDI signals. Based on previous study, the parameter values are assigned as described in Table 2 [16].

**2.3.3. Observation model for BOLD signals.** We utilize a typical hemodynamic model for BOLD signals [24, 25], which consists of neurovascular coupling, hemodynamics, and BOLD response. Neurovascular coupling describes how a vasoactive signal $s$ is generated from neuronal activity, which is described below:

$$s = \sum_n \beta_{n,exc} q_{n,exc} + \beta_{n,inh} q_{n,inh} + \beta_{n,ext} p_{n,ext}. \tag{7}$$

Here, $q_{n,exc}$ and $q_{n,inh}$ represent total inputs to a neural population $n$ from excitatory and inhibitory neural populations,

$$q_{n,exc} = \sum_{m \in exc} A_{nm} \sigma(v_m), \tag{8}$$

$$q_{n,inh} = \sum_{m \in inh} A_{nm} \sigma(v_m). \tag{9}$$

$p_{n,ext}$ denotes a weighted sum of direct external (stimulus) inputs to a neural population $n$,

$$p_{n,ext} = \sum_m C_{nm} u_m. \tag{10}$$

With vasoactive signal $s$, the standard hemodynamics model describes vasodilatory signal $h_1$, venous blood flow $h_2$, venous volume $h_3$, and deoxyhemoglobin $h_4$ as below:

$$\frac{dh_1}{dt} = s - \eta(h_1 - 1) - \chi(h_2 - 1), \tag{11}$$

$$\frac{dh_2}{dt} = h_1 - 1, \tag{12}$$

$$\frac{dh_3}{dt} = \left(h_2 - h_3^{1/\alpha}\right)/\tau, \tag{13}$$

$$\frac{dh_4}{dt} = \left(h_2\left(1 - (1-\varphi)^{\frac{1}{h_2}}\right)/\varphi - h_3^{\frac{1}{\alpha}}h_4/h_3\right)/\tau. \tag{14}$$

Here, $\eta$ and $\chi$ represent the rate of vasodilatory signal decay per sec and the rate of flow-dependent elimination; $\tau$ and $\alpha$ indicate rate of hemodynamic transit per sec and Grubb's exponent; $\varphi$ indicates resting oxygen extraction fraction.

The BOLD signal is generated by

$$Y_{BOLD} = g_{BOLD}(x, \phi) = V_0(k_1(1 - h_4) + k_2(1 - h_4/h_3) + k_3(1 - h_3)), \tag{15}$$

where $V_0$ indicates blood volume fraction; $k_1$ indicates intravascular coefficient; $k_2$ indicates concentration coefficient; $k_3$ indicates extravascular coefficient.

In summary, the goal of the current framework is to estimate unknown model parameters —both neural state model parameters (Table 1) and observation model parameters (Table 2)— using observed signals, a combination of multimodal data $[Y_{CaI}, Y_{VSDI}, Y_{BOLD}]$ from a single neural system. All parameters to be estimated in this study are detailed in Tables 1 and 2.

In the current multimodal framework, we used a shared neural state model $f(x,\theta)$ and combinations of observation models, i.e., $g_{CaI}(x, \phi_{CaI}), g_{VSDI}(x, \phi_{VSDI}), g_{BOLD}(x, \phi_{BOLD})$, to predict the observed multimodal signals, for example, $[Y_{CaI}, Y_{VSDI}, Y_{BOLD}]$.

## 2.4. Model parameter estimation using DCM

DCM estimates model parameters $\theta$ of model $m$ for observed signals $Y$ by maximizing the log-evidence ($\ln p(Y|m)$) [4,5], which is the sum of KL-divergence (between the posterior of the parameter $p(\theta|Y,m)$ and its approximate density $q(\theta)$) and the free energy $F$ evaluated under $q(\theta)$,

$$\ln p(Y|m) = KL(q(\theta)||p(\theta|Y, m)) + F(q(\theta), Y), \tag{16}$$

$$F(q(\theta), Y) = \int q(\theta) \ln \frac{p(Y, \theta)}{q(\theta)} d\theta. \tag{17}$$

The free energy $F$ serves an approximate lower bound for log evidence. Model optimization involves deriving an approximate density $q(\theta)$ that maximizes $F$, leading to $q(\theta) \approx [(\theta|Y,m)$. Under the Laplace assumption, $q(\theta)$ is iteratively estimated using the expectation (E) and maximization (M) steps of an Expectation-Maximization (EM) algorithm [5]. The free energy $F$ represents the sum of complexity and accuracy [5,26]. For details, see [4,5] and SPM12 (https://www.fil.ion.ucl.ac.uk/spm, code: spm_nlsi_GN.m).

To estimate model parameters from two or more signals, we modified the standard DCM model inversion to fit the multimodal data $[Y_{CaI}, Y_{VSDI}, Y_{BOLD}]$ from a single neural state model $f(x,\theta)$ with combinations of observation model $g_{CaI}(x, \phi_{CaI}), g_{VSDI}(x, \phi_{VSDI}), g_{BOLD}(x, \phi_{BOLD})$. Note that our approach incorporates free energy calculations using multiple observation signals. To facilitate this, we have modified **spm_int_ode.m** to accommodate the computation of all observation signals. These signals are then concatenated and processed through a free energy maximization step using the EM algorithm. This extension allows for a comprehensive analysis and integration of diverse data types within our framework.

Here, we can summarize this procedure with the function 'DCM', which calculates the maximized free energy, $F^*$, and the optimized posterior distribution, $q^*(\theta)$, i.e.,

$$(F^*, q^*(\theta)) = \text{DCM}(q(\theta), Y). \tag{18}$$

## 3. Method II: Construction of a ground truth foundational model for evaluating mms-DCM

### 3.1. Configuration of a foundational model for virtual systems

In the absence of a suitable multimodal experimental dataset acquired simultaneously from a single neural system, and the need for a ground truth to validate our proposed framework, we employed diverse virtual neural systems to generate CaI, VSDI, and fMRI signals. These virtual systems, serving as ground truth models for testing across various contexts, are modified versions of a foundational system. This foundational system is derived from real experimental data using computational modeling [1].

The foundational model was derived using CaI signals recorded from the barrel cortex of a mouse (animal ID: an194672) during a single whisker object localization task, as Peron [27] reported. We partitioned the observed spatiotemporal CaI data into four subregions to account for varying spatial coverages across different modalities. Using Independent Component Analysis (ICA), we then grouped cells within each subregion into two excitatory and one inhibitory neural populations, according to the conventional configuration of a cortical column in the Jansen-Rit model [8, 15], all showing immediate responses to stimuli. From these groupings, we extracted a total of 12 CaI signals—three from each subregion, corresponding to the two excitatory and one inhibitory neural populations identified. A more detailed description of this process can be found in S1 Text.

Extrinsic connections in the foundational model were established between subregions (here, we denote subregions as columns in Fig 1), while intrinsic connections were defined among neural populations within each subregion (column), as illustrated in Fig 2A. Similar to Jung et al. [1], we included a hidden external region, containing both inhibitory and excitatory populations, in the neural state model of the DCM to represent potential interactions between external unobserved neural populations and the observed neural populations of interest.

The parameters for the foundational model are estimated through a series of parameter estimation steps, which form a core component of the mms-DCM framework.

### 3.2. Model parameter configuration

In the DCM model, the biological parameters were expressed in exponential form, such as $X_i = X_{i0} \exp(\theta_i)$, to ensure the polarity of parameter, as in the conventional DCM method [25] (SPM code: spm_nlsi_GN.m). Here, $X_{i0}$ represents the reference value of a $i$-th biological parameter $X_i$, which is modulated by a parameter $\theta_i$. The goal of DCM parameter optimization (DCM function) was to search the optimal biological parameter $X_i$ by estimating the modulation parameter $\theta_i$ using a prior distribution of $\theta_i \sim N(0, 1/32)$ for a given reference value $X_{i0}$. By

assigning a zero-mean prior, the real biological property $X_i$ was allowed to vary around the reference value $X_{i0}$. In the current study, we used modulation parameters to represent effective connectivity $A$, input modulation parameter $C$, the decay time constant $T$ in the neural state model, the conversion ratio $k_{Ca}$ from calcium ion current to concentration per unit time, and the time constant $\tau_{Ca}$ for calcium decay, as described in Tables 1 and 2.

### 3.3. Bayesian optimization and DCM for model parameter estimation

Due to the exponential parameterization, model parameter estimation requires determining both a reference value and its modulation parameter. The reference value is first established, and then the modulation parameter is fine-tuned, rather than estimating both simultaneously. In a conventional optimization setting, the reference value functions as a hyperparameter. To enhance the efficiency of parameter estimation, we employed Bayesian optimization (referred to as BAYESopt, distinct from DCM in Eq 18) [28] to determine the reference values. These reference values then serve as priors for fine-tuning DCM parameter estimation (Eq 18), facilitating an effective fit to the experimental data.

In BAYESopt, a type of Bayesian optimization, we defined a cost function, DCMopt, which returns maximized free energy derived from DCM for given priors $q(\theta)$ and data Y, based on Eq 18, as below:

$$F^* = \text{DCMopt}(q(\theta), Y)$$

DCMopt provides free energy by inferring the neural state model model $f(x,\theta)$ and integrating it with combinations of observation models such as $g_{Cal}(x, \phi_{Cal}), g_{VSDI}(x, \phi_{VSDI}), g_{BOLD}(x, \phi_{BOLD})$ for a given data set and parameter set $\theta$.

Among Bayesian optimization algorithms, we implemented a Gaussian process model as the surrogate model [28] and used an expected-improvement-per-second-plus function for the acquisition function [29,30]. This approach provides a point estimate, while DCMopt, using the variational Laplace scheme, estimates model parameters in terms of posterior distribution, as explained in the previous section.

To improve computational efficiency, DCMopt is divided into 'fast' and 'precise' modes based on the number of EM cycle iterations in DCM: Fast DCMopt runs a single cycle, whereas precise DCMopt runs under a convergence condition of the maximum number of iterations 128, and the free energy gradient threshold $10^{-2}$. Initially, fast DCMopt is applied in BAYESopt, with precise DCMopt employed subsequently for more thorough analysis.

Both BAYESopt and DCM are instrumental in optimizing modulation parameters but differ in their specifics. BAYESopt primarily focuses on finding the expectations of modulation parameters, $\bar{\theta}_i$. In contrast, DCM searches for posteriors (expectations and variances) of modulation parameters constrained by Gaussian prior distributions, denoted as $N(\bar{\theta}_i, \sigma_i^2)$, where $\bar{\theta}_i$ and $\sigma_i^2$ represent the expectation and variance of Gaussian distribution for the $i$-th parameter. The prior variation $\sigma_i^2$ for DCMopt is assigned a predetermined value as described in Table 1. The results from BAYESopt are used to set prior expectations for fine-tuning in DCM. Consequently, BAYESopt serves as a hyperparameter optimization tool, considering parameter expectations $\bar{\theta}_i$ (not distributions) as hyperparameters and using DCMopt as a cost function, as detailed in Box 1.

### 3.4. Parameter estimation for the foundational model

We implemented the described parameter estimation method on the experimental CaI signals using the foundational circuit model depicted in Fig 2A. We observed consistent patterns of intra-columnar connectivity across all columns (from column 1 to column 4), as illustrated in Fig 2B.

---

### Box 1. Pseudo-algorithm for Bayesian parameter optimization and DCM optimization

% For observation signals Y, the prior distribution of a parameter $X$, $q(\theta_X|\bar{\theta}_X, \sigma_X^2)$, follows a Gaussian distribution, and a parameter $X$ is represented in the form of $X = X_0 \exp(\theta_X), \theta_X \sim N(\bar{\theta}_X, \sigma_X^2)$.

function $F^* = \text{DCMopt}(q(\theta|\bar{\theta}, \sigma_\theta^2), Y, \text{max\_iteration})$

 % DCM algorithm explores optimal parameter that maximizes the free energy.

 $(F^*, q(\theta|\bar{\theta}, \sigma_\theta^2)) = \text{DCM}(q(\theta|\bar{\theta}, \sigma_\theta^2), Y, \text{max\_iteration})$

end

function $[\bar{\theta}] = \text{BAYESopt}(\bar{\theta}, Y, \text{SR}(\bar{\theta}))$

 % Exploring the optimal parameter expectation $\bar{\theta}$ for given data Y

 % within the given search range $\text{SR}(\bar{\theta})$.

 % Fast search

 for i = 1: $N_{fast}$

 $\bar{\theta} = \text{BAYES\_OPTIMIZATION\_UNIT}(\text{DCMopt}(q(\theta|\bar{\theta}, \sigma_\theta^2), Y, 1), \text{SR}(\bar{\theta}))$

 end

 % Precise search within the narrowed search range $\text{SR}_{prcise}(\bar{\theta})$

 for i = 1: $N_{precise}$

 $\bar{\theta} = \text{BAYES\_OPTIMIZATION\_UNIT}(\text{DCMopt}(q(\theta|\bar{\theta}, \sigma_\theta^2), Y, 128), \text{SR}_{prcise}(\bar{\theta}))$

 end

end

---

For instance, in every column, there was pronounced excitatory connectivity from E#2 to I#1, where 'E' and 'I' denote excitatory and inhibitory neuronal populations, respectively, '#' indicates the column number, and the last number identifies the population. Conversely, the connectivity from I#1 to E#1 was notably weak. Note that these parameters were estimated from experimental data obtained from a mouse. Although the accuracy of the model parameters has not been validated, we considered these parameters as a ground truth reference for subsequent virtual experiments by simulating diverse signals. The simulation results, utilizing the estimated parameters, are displayed in Fig 2C, and the details of the parameter optimization are outlined in Table 1.

We constructed diverse virtual models based on the estimated foundational model and generated a range of signals across various virtual experimental settings to validate our mms-DCM scheme.

## 4. Results: Experiments

Utilizing the virtual neural state model, we generated CaI ($Y_{\text{CaI}}$), VSDI ($Y_{\text{VSDI}}$), and BOLD ($Y_{\text{BOLD}}$) signals at specific regions with varying spatial scales, as depicted in Fig 1, to simulate diverse experimental scenarios.

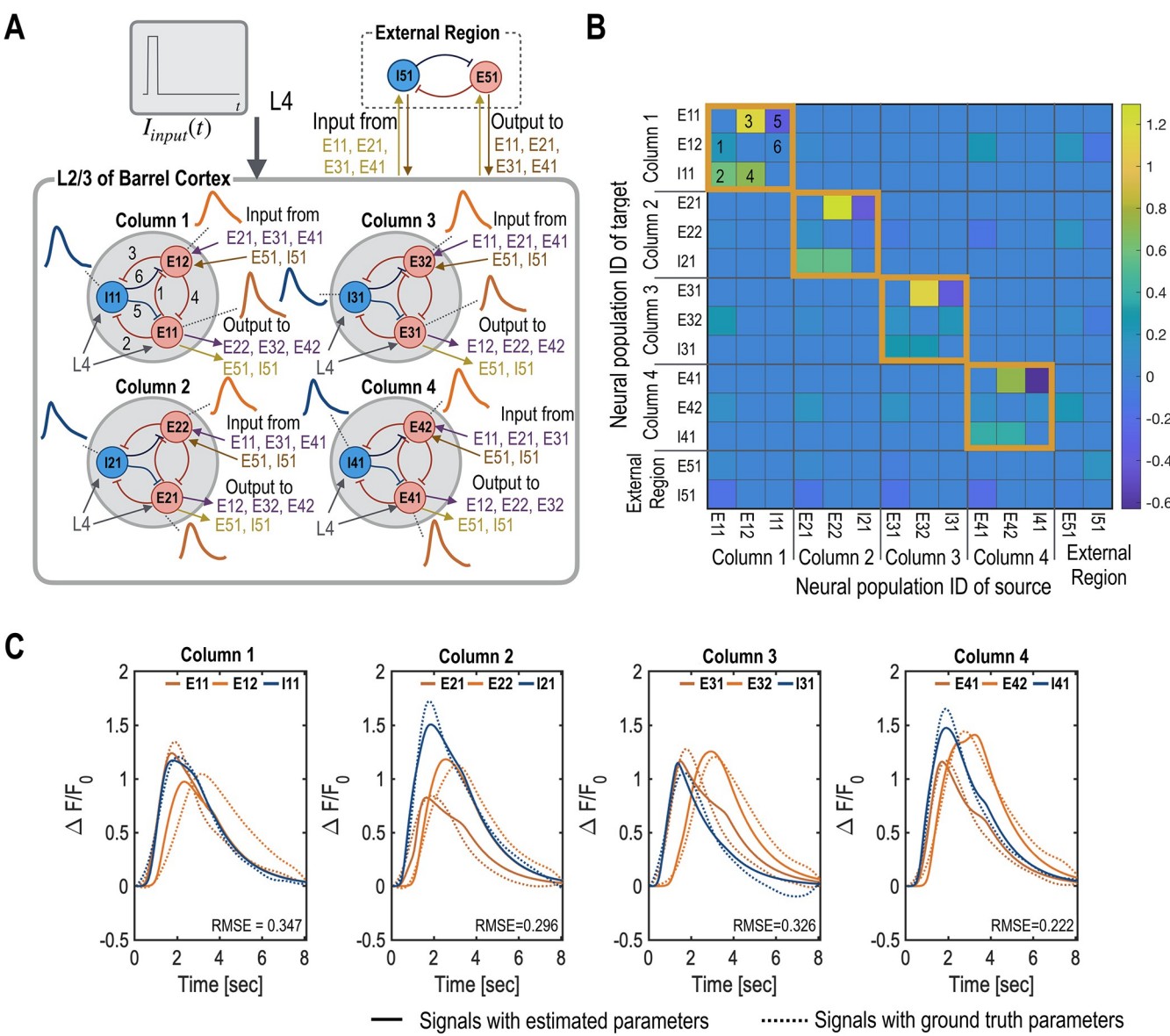

**Fig 2. The ground truth foundational model for mms-DCM experiments.** The ground truth foundational model for mms-DCM experiments is constructed based on CaI signals from layer 2/3 (L2/3) of the mouse barrel cortex. For this model, we extracted 12 CaI signals corresponding to four subregions (denoted as columns in the current study), each comprising three neural populations: two excitatory and one inhibitory. A. The virtual circuit model, illustrating both intra-columnal and inter-columnal connections, is displayed. We assume identical connections across each column. B. The estimated effective connectivity is shown, highlighting the dynamic interactions between neural populations. C. The reproduction of CaI signals, based on the estimated circuit parameters, is depicted. Solid lines represent the generated signals, contrasting with the dotted lines, which correspond to the experimental data. The different colors—red, orange, and blue—represent the excitatory populations E#1, E#2, and the inhibitory population I#1, respectively, with '#' indicating the column number.

To validate mms-DCM for model estimation of a partially observed multiscale system, we conducted five experiments with six simulations, each incorporating different combinations of observation signals.

1. **Experiment 1**: We evaluated the performance of the proposed iterative estimation scheme for integrating multimodal signals in the parameter estimation process of mms-DCM.

2. **Experiment 2**: Focused on demonstrating the advantages of utilizing both local and global circuit information in parameter estimation of the partially observed system:

   a. **Experiment 2a**: Showcased the utility of using a local circuit's prior information for estimating global circuit parameters.

   b. **Experiment 2b**: Illustrated the effectiveness of employing a global circuit's prior information for estimating local circuit parameters.

3. **Experiment 3**: We constructed a scenario where multimodal signals were acquired from multiple independent experiments conducted at different times on the same neural circuit system. This experiment assessed the multimodal integration scheme's capability to handle asynchronous data acquisition and within-subject variations in a neural system.

4. **Experiment 4**: This experiment involved the integration of all multimodal signals––CaI, VSDI, and BOLD––in the estimation of neural circuitry at a region with partially observed signals. We tested the performance of mms-DCM in modeling a neural circuit using signals of different scales.

5. **Experiment 5**: This experiment aimed to showcase the applicability of mms-DCM to a more complex system extending to four regions. In this experiment, we modeled a larger circuit with CaI signals at a column, VSDI at two columns within a region, and BOLD signals across four regions.

All these experiments are summarized in Table 3.

## 4.1 Experiment 1: iterative parameter estimation of multiscale circuit in mms-DCM

We expand upon the model parameter estimation steps described in 3.1.2 and introduce an iterative scheme for parameter estimation in multiscale partially observed systems. We also evaluate the proposed scheme using virtually generated data.

For the evaluation, we partially generated three CaI signals for neural populations exclusively in column 1 and four VSDI signals across columns 1 to 4, with one VSDI signal per column. These signals were elicited by applying three distinct inputs to the virtual model with intensities of 1.0, 0.8, and 1.2 mA for 1.1 seconds within an 8-second interval, as depicted in Fig 3.

The parameter estimation process, illustrated in Fig 4, is an iterative procedure that reciprocally leverages local and global circuit information.

In step 1, the process begins with estimating local circuit parameters within column 1 using the three available CaI signals. The estimated parameters serve as priors for the parameters in columns 2, 3, and 4, where CaI signals are not available. In this step, BAYESopt is employed to estimate three modulation parameters corresponding to intra-regional connectivity ($A_{intra}$), input modulation strength ($C$), and the time constant ($T$).

In step 2, we focus on optimizing the global circuit parameters by refining the prior expectations of modulation parameters for inter-regional connectivity ($\bar{\theta}_{A_{inter}}^{step2}$). The posterior expectations of local circuit parameters (e.g., $\bar{\theta}_{A_{intra}}^{step1}$) from column 1 (found in step 1) serve as prior expectations for local parameters in the other columns lacking CaI signals. The optimal parameter expectations from this step 2 ($\bar{\theta}_{A_{inter}}^{step2}$) define the search ranges (SR) for step 3.

Step 3 updates modulation parameters for the local circuit, i.e., $A_{intra}$, $C$, and $T$. The BAYESopt search ranges are set within ±30% from the optimal parameter expectations estimated in step 2 ($\bar{\theta}_i^{step3} \in [0.7 \times \bar{\theta}_i^{step2}, 1.3 \times \bar{\theta}_i^{step2}]$).

**Table 3. Summary of experiments.**

| | Description | Observation signals | Figure |
|---|---|---|---|
| **Estimation** | Construction of the ground truth system | 12 CaI signals ($Y_{CaI;c1,c2,c3,c4}$) | Fig 2 |
| **Experiment 1** | Iterative model estimation using CaI and VSDI | 3 CaI signals at column 1 ($Y_{CaI;c1}$), 4 VSDI signals at 4 columns ($Y_{VSDI;c1,c2,c3,c4}$) | Figs 3 and 4 |
| **Experiment 2a** | Utility of using **local circuit information** in global circuit estimation using CaI and VSDI | **Method 1 & 2**: 3 CaI signals at column 1 ($Y_{CaI;c1}$), 4 VSDI signals at 4 columns ($Y_{VSDI;c1,c2,c3,c4}$) **Method 3**: 4 VSDI signals at 4 columns ($Y_{VSDI;c1,c2,c3,c4}$) | Fig 5 |
| **Experiment 2b** | Utility of using **global circuit information** in local circuit estimation using CaI and VSDI | **Method 1**: 3 CaI signals at column 1 ($Y_{CaI;c1}$), 1 VSDI signal at column 1 ($Y_{VSDI;c1}$) **Method 2**: 3 CaI signals at column 1 ($Y_{CaI;c1}$), 4 VSDI signals at 4 columns ($Y_{VSDI;c1,c2,c3,c4}$) | Fig 6 |
| **Experiment 3** | mms-DCM for CaI and VSDI sampled at different time experiments | **Case 1**: 3 CaI signals at column 1 obtained with perturbed $C$ ($Y'_{CaI;c1}$), 4 VSDI signals at 4 columns ($Y_{VSDI;c1,c2,c3,c4}$) **Case 2**: 3 CaI signals at column 1 ($Y_{CaI;c1}$), 4 VSDI signals at 4 columns obtained with perturbed $C$ ($Y'_{VSDI;c1,c2,c3,c4}$) | Fig 7 |
| **Experiment 4** | mms-DCM with all multimodal signals of CaI, VSDI and BOLD. | 3 CaI signals at column 1 ($Y_{CaI;c1}$), 2 VSDI signals at columns 1 and 2 ($Y_{VSDI;c1,c2}$), 2 BOLD signals at regions r1 and r2 ($Y_{BOLD;r1,r2}$). | Fig 8 |
| **Experiment 5** | mms-DCM of CaI, VSDI and BOLD for an extended system (four regions) | 3 CaI signals at column 1 ($Y_{CaI;c1}$), 2 VSDI signals at columns 1 and 2 ($Y_{VSDI;c1,c2}$), 4 BOLD signals at regions $r1$, $r2$, $r3$, and $r4$ ($Y_{BOLD;r1,r2,r3,r4}$). | Fig 9 |

[1] In **Experiments 1 to 4**, there are a total of 14 neural populations, which include two neural populations in the hidden external region.

[2] In **Experiment 5**, the number of neural populations is 28, encompassing four hidden external neural populations.

[3] The indices $c1$, $c2$, $c3$, and $c4$ denote columns. For instance, the observed CaI signals in column 1 are represented as $Y_{CaI;c1}$, and include two excitatory neural populations (E11 and E12) and one inhibitory neural population (I11).

[4] The indices $r1$, $r2$, $r3$, and $r4$ denote regions. For example, the observed BOLD signals in region 1, which spans two columns, are represented as $Y_{BOLD;r1}$.

[5] All experiments employed the same strategy for parameter estimations. Details of the estimation steps are displayed in the corresponding figures.

In step 4, we perform a fine-tuning of model parameters based on the comprehensive information gained from both local and global circuit estimations. The expectations of modulation parameters for $A_{inter}$, $A_{intra}$, $C$, and $T$ are explored within a search range of $\pm 10\%$ of the best values estimated in step 3 ($\bar{\theta}_i^{step4} \in [0.9 \times \bar{\theta}_i^{step3}, 1.1 \times \bar{\theta}_i^{step3}]$). Note that expectations of modulation parameters for $A_{intra}$, C, T of column 1, which uniquely have CaI signals, are utilized across all columns in steps 3 and 4. From steps 2 to 4, we iteratively proceeded with each step until the optimized free energy exceeded the previous maximum.

The final parameter estimation is conducted using DCM with a maximum number of iterations 128, and the free energy gradient threshold $10^{-2}$.

To evaluate the effectiveness of the proposed iterative parameter estimation scheme, we compared it with a conventional estimation procedure where all the parameter expectations of local and global circuits are searched simultaneously, referred to as 'one-step estimation' as opposed to iterative estimation.

**Estimation results.** As shown in Fig 4B–4E, iterative estimation methods yield more accurate results than the one-step method. The iterative approach produced a closer reproduction of CaI and VSDI to the ground truth, which can be quantitatively assessed by the root

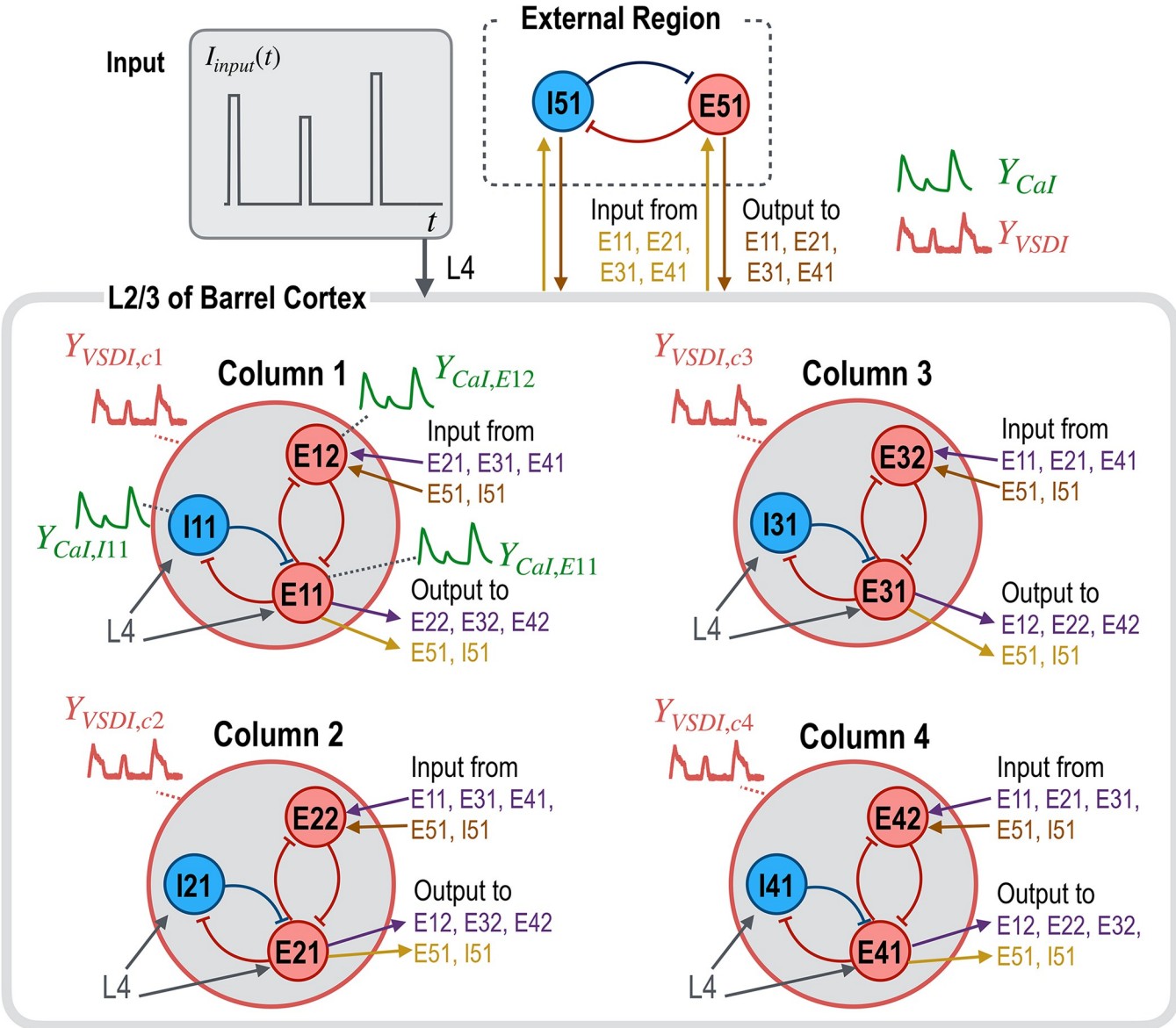

**Fig 3. Multimodal signal generation using a virtual neural circuit model.** We used a virtual neural circuit model to selectively generate multimodal signals. This model serves as a ground truth and features four columns alongside a hidden external region, each column designed with identical connection topologies that include two excitatory and one inhibitory neural population. To add complexity to the model, we allowed for slight variations in the strength of intra-regional intrinsic connectivity among these columns. The strengths of these connections were derived from experimental data, specifically CaI signals (refer to Fig 2). For our simulations, CaI signals were generated exclusively for neural populations in column 1, while four VSDI signals were produced for all four columns to demonstrate the broader spatial coverage of this modality. Additionally, the hidden external region, which contains both inhibitory and excitatory neural populations, is included in the model but omitted in subsequent figures for simplicity. This selective signal generation strategy aims to reflect the diverse imaging capabilities of each signal modality within the intricate neural circuitry of the virtual model.

mean square error (RMSE). Specifically, the RMSE values from iterative estimation were 0.194 for CaI and 0.142 for VSDI, significantly lower than those from one-step estimation, which were 0.639 for CaI and 0.433 for VSDI.

Moreover, the correlation coefficient between the estimated parameters and the ground truth parameters is higher in the iterative estimation ($r = 0.623$) compared to the one-step estimation ($r = 0.299$).

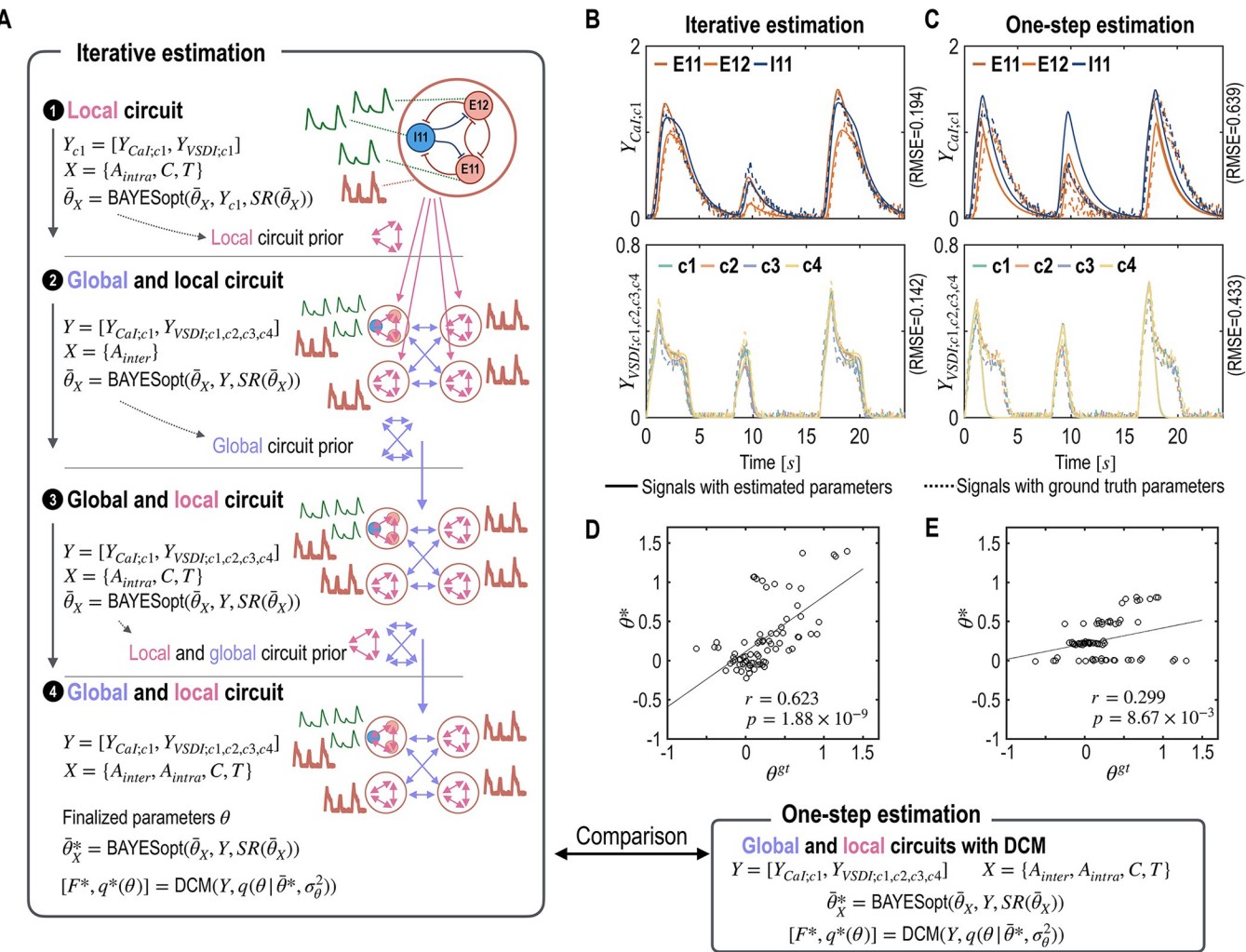

**Fig 4. Iterative parameter estimation of local and global circuit parameters for multimodal signals.** A. This figure outlines an iterative process for estimating neural circuits based on CaI and VSDI signals. Initially, local circuit parameters are estimated using available CaI signals from column 1. These estimated local parameters serve as expectations of prior distributions for parameters in other columns, facilitating the exploration of inter-regional interaction parameters ($A_{inter}$). The third step updates expectations of prior distributions for intra-regional connectivity parameters, while the final step precisely refines those of both global and local circuit parameters. B and C. Simulated signals from the ground truth (dotted lines) and from the estimated parameters (solid lines) are displayed. D and E. Comparisons between the ground truth parameters ($\theta^{gt}$, x-axis) and the estimated parameters ($\theta^*$, y-axis) are presented for both iterative (D) and one-step (E) estimations. The results from the optimal model parameters in both iterative and one-step estimations are compared in (B) ~ (E).

## 4.2 Experiment 2: mms-DCM utilizes reciprocal information for estimating local and global circuit parameters

### 4.2.1 Experiment 2a: local circuit information improves global circuit estimation.
To evaluate the impact of local circuit information on global circuit parameter estimation, we compared three different methods illustrated in Fig 5.

Method 1 involves the first two steps of Experiment 1, utilizing local circuit priors for global circuit estimation. In Step 1, the local circuit is estimated using 1 VSDI and 3 CaI signals from column 1 ($Y_{method1;step1} = [Y_{CaI;c1}, Y_{VSDI;c1}]$). The estimated local circuit posteriors from column 1 are then used as priors for each column in Step 2. In this step, 3 CaI signals from column 1, along with an additional 4 VSDI signals covering all columns, are employed to fit all circuit parameters ($Y_{method1;step2} = [Y_{CaI;c1}, Y_{VSDI;c1,c2,c3,c4}]$).

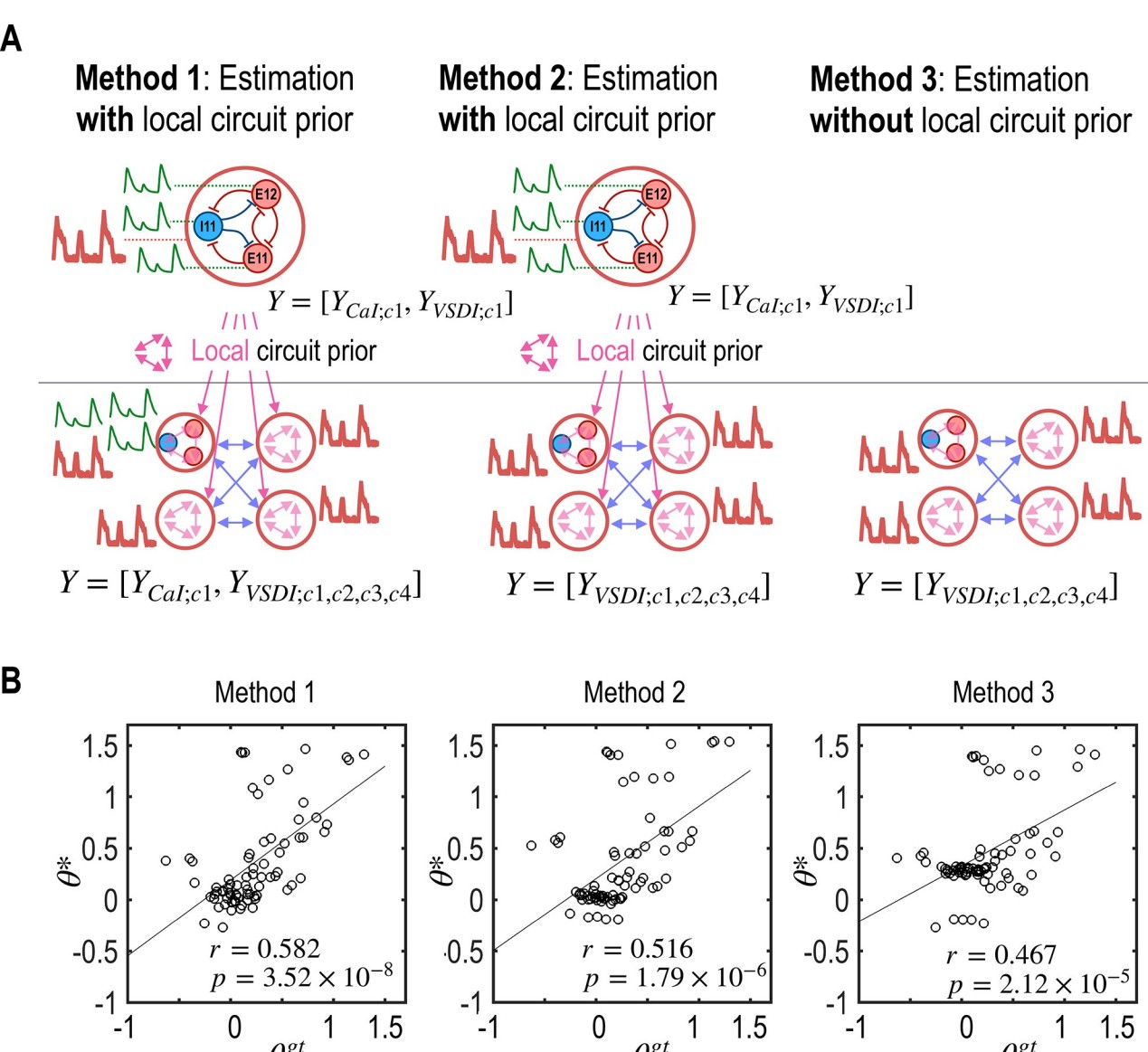

**Fig 5. Evaluation of global circuit parameter estimation using local circuit priors (Experiment 2a).** A. Three different methods, designated as Method 1, Method 2, and Method 3, are depicted schematically. B. Scatter plots show correlations between the estimated parameters ($\theta^*$) and the ground truth parameters ($\theta^{gt}$). Method 1, which incorporates local circuit information into global circuit estimation, demonstrates the highest accuracy, showcasing the significant impact of utilizing local data to enhance global parameter estimation accuracy.

Method 2 mirrors Method 1's conditions by using local parameter posteriors from column 1 for other columns. However, this method uses only four VSDI signals for model fitting in Step 2, excluding the local CaI signals from column 1 ($Y_{\text{method2};step1} = [Y_{CaI;c1}, Y_{VSDI;c1}]$, $Y_{\text{method2};step2} = [Y_{VSDI;c1,c2,c3,c4}]$).

Method 3 utilizes only the four VSDI signals, without incorporating any local information estimated from CaI signals, to estimate global circuit inter-regional connectivity ($Y_{\text{method3}} = [Y_{VSDI;r1,r2,r3,r4}]$).

The results, presented in Fig 5, show that the accuracy of parameter estimation is significantly influenced by the inclusion of local circuit information. Method 1, which integrates both local CaI and VSDI signals along with the posteriors of CaI signal estimation across

columns, achieved the highest correlation with the ground truth parameters ($r = 0.582$, $p = 3.52 \times 10^{-8}$). Method 2, using the same initial local circuit posteriors but omitting CaI signals for fine-tuning in the second step, showed a lower correlation ($r = 0.516$, $p = 1.79 \times 10^{-6}$). Method 3, relying only on VSDI signals, demonstrated the lowest correlation ($r = 0.467$, $p = 2.12 \times 10^{-5}$).

These findings underscore the significance of integrating local circuit information to accurately capture the dynamics of global neural circuits. The results affirm the importance of combining multimodal, multiscale signals for more precise global circuit model estimation.

**4.2.2 Experiment 2b: global circuit information supports local circuit estimation.** To assess the impact of global circuit information on local circuit dynamics, we compared local circuit parameters derived under two different contexts, as depicted in Fig 6. In Method 1 (local parameter estimation without global circuit information), the focus was solely on the local circuit of column 1. This method isolated the column from others, using only CaI signals from this column for parameter estimation. In contrast, Method 2 incorporated both local and global circuit information to fit the local CaI signals at column 1 along with the global VSDI signals from all columns.

The correlation of the ground truth with models' estimated parameters for the local circuit with global circuit information (Method 2) was $r = 0.692$ ($p = 0.018$) while those without global circuit information (Method 1) was $r = 0.530$ ($p = 0.094$) (Fig 6). These results highlight the significant role of global circuit information in achieving accurate estimations of local circuit dynamics. It underscores the interdependence between local and global neural activities, demonstrating that global insights can enhance the precision of local parameter estimations.

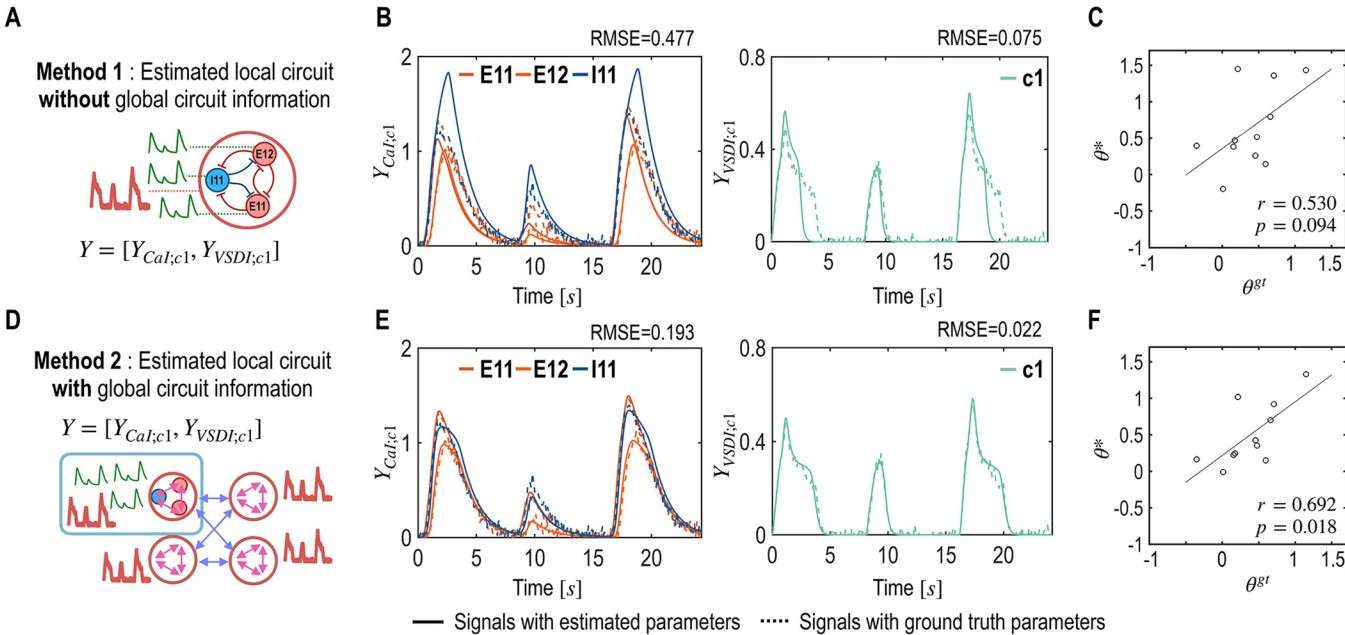

**Fig 6. Utility of the global circuit information on local circuit estimation (Experiment 2b)**. A. This experiment compares the local circuit of column 1 under two different evaluation contexts. In Method 1, the local circuit is evaluated in isolation (without global circuit information), focusing solely on its inherent dynamics. B. Estimated and ground truth CaI and VSDI signals of Method 1 are plotted. The solid lines in the plots are generated from the estimated parameters, reflecting the model's interpretation of the neural activity. Conversely, the dotted lines represent the signals derived from the ground truth parameters, serving as a benchmark for evaluating the accuracy of the model's estimations. C. Scatter plots of Method 1 show correlations between the estimated parameters ($\theta^*$) and the ground truth parameters ($\theta^{gt}$) for Method 1. D. Method 2 evaluates the same local circuit within a broader global context, incorporating global circuit information to potentially enhance parameter estimation accuracy. E. CaI and VSDI signals of Method 2 are plotted. F. Scatter plot between the estimated parameters and the ground truth parameters of Method 2 is displayed.

### 4.3 Experiment 3: mms-DCM for integrating multimodal data across experiments

In practical experimental setups, acquiring multimodal signals simultaneously can be challenging. A more feasible alternative involves repeating experiments to capture data from different modalities separately. This experiment evaluates the utility of the mms-DCM framework, considering that data are collected from two separate experiments at slightly different time points from the same neural system. The experiment operates under the assumption that while dynamic neural activities may vary, the most critical system parameters, such as connectivity and intrinsic properties, remain relatively consistent.

To simulate the collection of multimodal data at different times, we introduced slight variations in the system's input gains across the time points. This was achieved by modifying the input gain parameter $C$, perturbing it with noise following a normal distribution $N(0, 0.04)$. This experiment aimed to mimic the variability encountered in real-world data acquisition.

We created two scenarios: In the first, we combined the original VSDI signals with CaI signals of the perturbed system, represented as $Y = [Y'_{CaI;r1}, Y_{VSDI;r1,r2,r3,r4}]$. In the second scenario, we used perturbed VSDI signals alongside the original CaI signals, denoted as $Y = [Y_{CaI;r1}, Y'_{VSDI;r1,r3,r3,r4}]$, where $Y'$ indicates signals from the perturbed system (Fig 7A and 7D). We then evaluated the accuracy of parameter estimation using the mms-DCM, following a procedure similar to that in Experiment 1.

The analysis yielded moderate fitting results for the perturbed signals (Fig 7B and 7E). The correlation with ground truth parameters for the perturbed CaI signals was $r = 0.460$ ($p = 2.88 \times 10^{-5}$), and for the perturbed VSDI signals, the correlation was $r = 0.350$ ($p = 0.002$) (Fig 7C and 7F). These findings indicate that despite the introduction of noise into the input

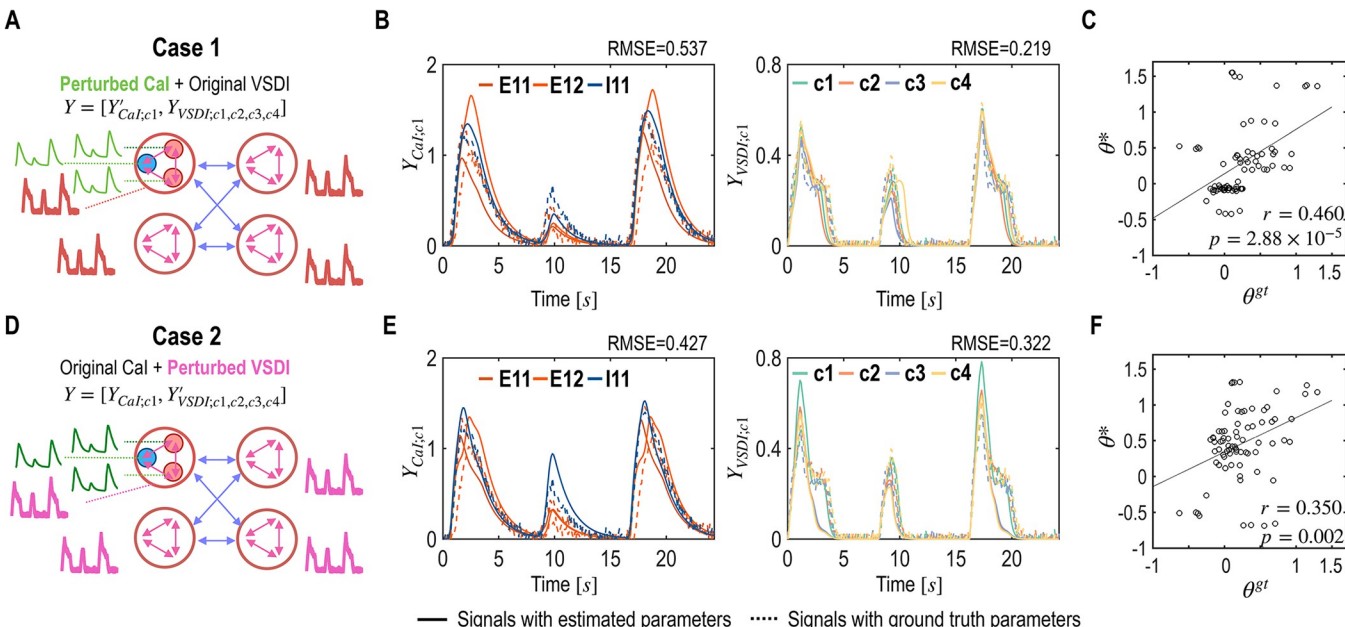

**Fig 7. Results of mms-DCM estimation with integrated multimodal data across replicated experiments (Experiment 3).** A. Two cases from the experiment are schematically displayed. In Case 1, CaI signals were obtained from a perturbed system, reflecting intentional variations in input parameters to simulate real-world data inconsistencies. B. Estimated and ground truth CaI and VSDI signals of Case 1 are plotted. The solid lines are generated from the estimated parameters, while the dotted lines depict signals derived from the ground truth parameters. C. Scatter plot of Case 1 shows correlations between the estimated parameters ($\theta^*$) and the ground truth parameters ($\theta^{gt}$). D. In Case 2, we used perturbed VSDI signals, introducing similar variability. E. Estimated and ground truth CaI and VSDI signals of Case 2 are plotted. F. Scatter plot of the estimated and ground truth parameters of Case 2 is displayed.

parameters, the mms-DCM method can maintain a reasonable level of accuracy in estimating model parameters, particularly concerning the neural circuit.

## 4.4 Experiment 4: mms-DCM for integration of multiscale multimodal CaI, VSDI, and BOLD signals

In this section, we expand the mms-DCM framework to encompass BOLD signals alongside the existing CaI and VSDI signals. In the present configuration, the VSDI signals from columns 3 and 4, used in the previous section, are omitted. Instead, BOLD signals have been gathered from two broader regions: r1, which encompasses both columns 1 and 2, and r2, which comprises columns 3 and 4; $Y = [Y_{CaI;c1}, Y_{VSDI;c1,c2}, Y_{BOLD;r1,r2}]$. Similar to the previous section, we apply an iterative estimation methodology to manage multiscale observed signals ([Fig 8A]). By integrating these BOLD signals from combined columns, we aim to enhance the estimation process by leveraging the distinct temporal and spatial characteristics of BOLD responses relative to the more localized and direct measurements of neuronal activity obtained from VSDI and CaI signals.

We also compare the results of iterative estimation with those of one-step estimation as in section 4.1. The observation signals of CaI, VSDI, and BOLD were better fitted in the iterative estimation, compared to the one-step estimation ([Fig 8B and 8C]). For the iterative estimation, the RMSE for CaI, VSDI, and BOLD were 0.561, 0.108, and 2.497, respectively ([Fig 8B]), which were lower than those from the one-step estimation, which were 0.655, 0.219, and 10.673 ([Fig 8C]). Correlations between the estimated model parameters and ground truth were also higher in the iterative estimation ($r = 0.340$, $p = 0.003$) than in one-step estimation ($r = 0.138$, $p = 0.236$) ([Fig 8D and 8E]). This result underscores the effectiveness of iterative estimation in capturing the complex dynamics of neural circuits across multiple observational modalities and demonstrates the effectiveness of mms-DCM with CaI, VSDI, and BOLD signals.

## 4.5 Experiment 5: mms-DCM for an extended system

In this section, we extend the application of mms-DCM to a more complex system consisting of two greater *Regions*, as illustrated in [Fig 9A]. Here, a *Region* includes two regions (or four columns). We hypothesize that these *Regions* are identical and interconnected through an interaction from column 2 of the first *Region* to column 1 of the second *Region*. The observational signals used for analysis are as follows: for the first *Region*, we use the same set as in the previous section ($Y = [Y_{CaI;c1}, Y_{VSDI;c1,c2}, Y_{BOLD;r1,r2}]$), which includes 3 CaI at region 1 ($Y_{CaI;c1}$), 2 VSDI at region 1 ($Y_{VSDI,c1,c2}$), and 2 BOLD signals at region 1 and 2 ($Y_{BOLD;r1,r2}$). The second *Region* is analyzed using only two BOLD signals ($Y_{BOLD;r3,r4}$).

Following the iterative mms-DCM estimation procedure described in section 4.4, we first perform an iterative estimation for the first barrel cortex column. The estimated posterior neural circuit parameters of column 1 are then used as priors for the circuit estimation of column 2 (as shown in [Fig 9A]). This approach ensures a continuous optimization of both global and local circuit parameters, including the inter-columnar interactions. In the final step, we select the optimal solution based on the highest free energy. This result is compared with a one-step estimation, as done in the previous sections.

The experiment results are consistent with trends observed in the previous sections: while the one-step estimation produces a moderate level of signal reproduction, the iterative process exhibits enhanced performance. Particularly notable is the one-step estimation's significant shortcoming in accurately reproducing BOLD signals for the second column ([Fig 9B and 9C]). For the iterative estimation, the RMSE for CaI, VSDI, and BOLD were 0.213, 0.057, and 4.355, respectively ([Fig 9B]), which were lower than those from the one-step estimation, which were

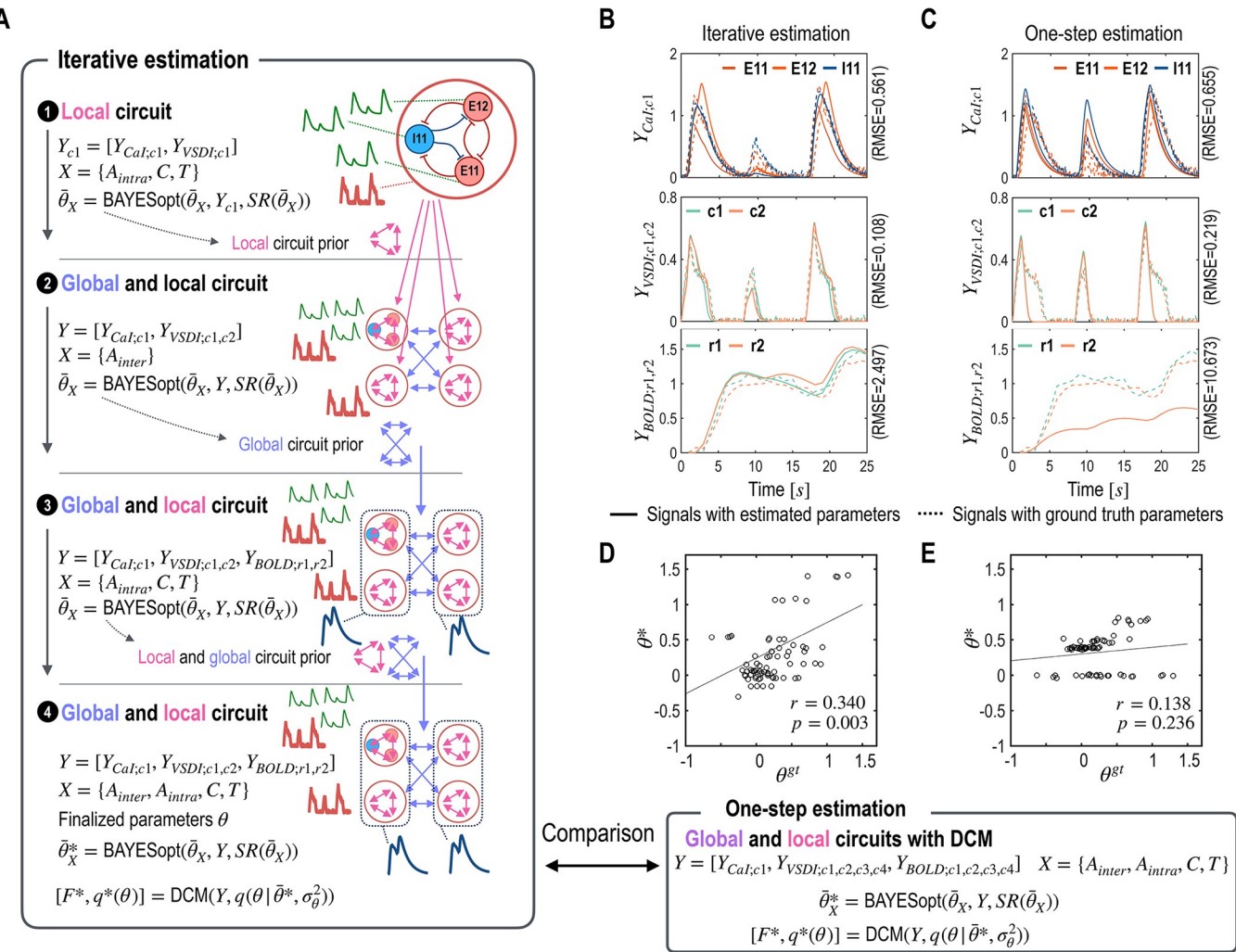

**Fig 8. Parameter estimation incorporating local and global circuit priors for multimodal observation signals; CaI, VSDI, and BOLD signals.** A. This outlines an iterative process for estimating neural circuits based on CaI, VSDI, and BOLD signals. The process begins by estimating the local circuit parameters using CaI signals available exclusively in column 1. These estimated local parameters then serve as prior expectations for parameters in other columns, facilitating the exploration of inter-regional interaction parameters ($A_{inter}$). The third step updates expectations of prior distributions for intra-regional connectivity parameters, and the final step meticulously refines those of both global and local circuit parameters, ensuring comprehensive integration of all data sources. B and C. Simulated signals (CaI, VSDI, and BOLD) from the ground truth (dotted lines) and the estimated parameters (solid lines) are displayed. D and E. Correlations between the ground truth parameters ($\theta^{gt}$, x-axis) and the estimated parameters ($\theta^*$, y-axis) are presented for the iterative (D) and the one-step (E) estimation schemes. The simulation results with optimal model parameters obtained from the iterative and the one-step estimation schemes are compared in (B) ~ (E).

0.641, 0.218, and 17.962 (Fig 9C). Additionally, the correlation between the ground truth and the estimated parameters from the iterative estimation (r = 0.411, p = 1.16 ×10$^{-7}$) is substantially higher than that from the one-step estimation (r = 0.101, p = 0.212), reinforcing the effectiveness and increased accuracy of the iterative mms-DCM approach when applied to a more complex neural system (Fig 9D and 9E).

To evaluate how effectively our estimation methods refined parameter estimates by reducing uncertainty, we calculated the posterior shrinkage, as described in [31], defined by:

$$\rho_i = 1 - \frac{\sigma^2_{i,post}}{\sigma^2_{i,prior}},$$

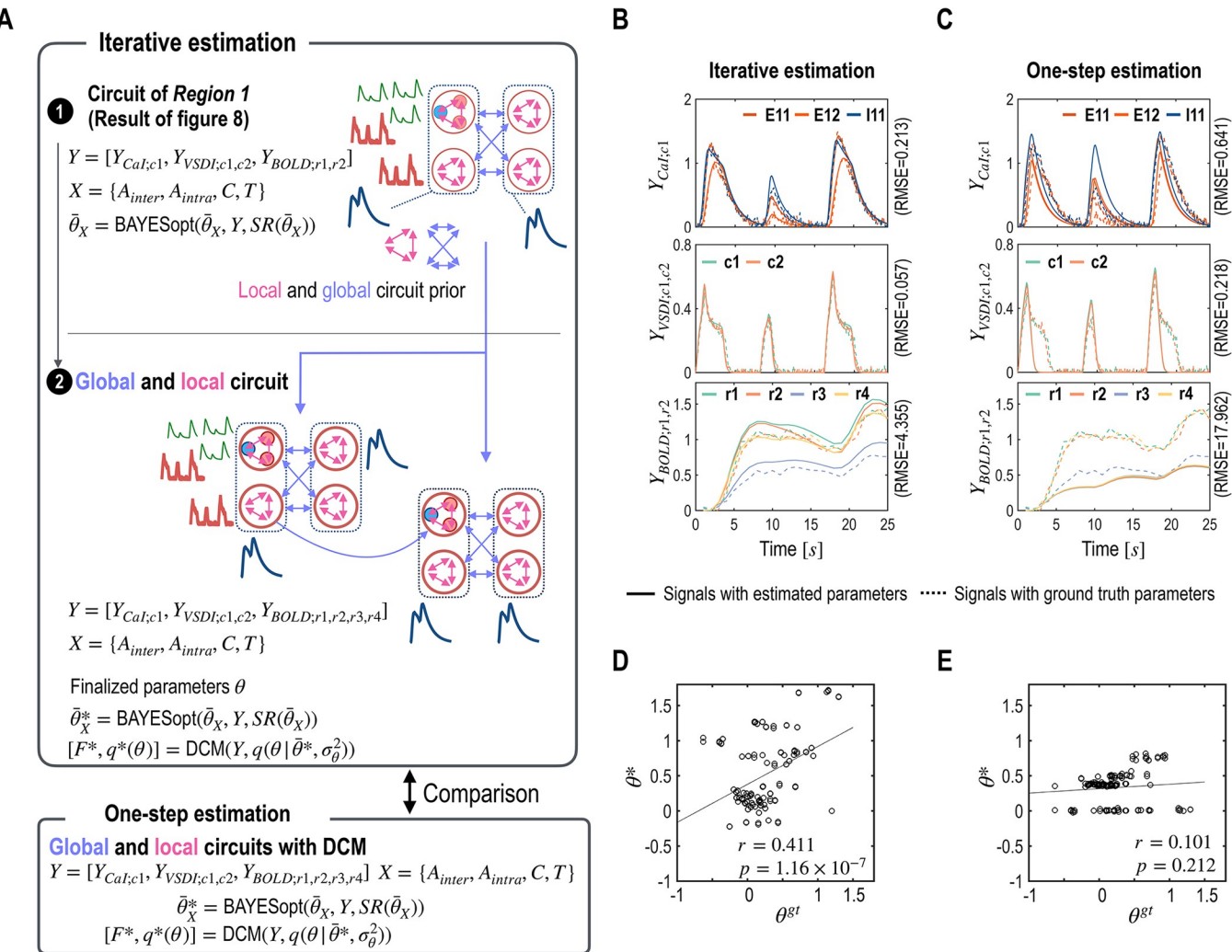

**Fig 9. Results of iterative mms-DCM for extended system that consists of two *Regions*.** A. The estimation process for the extended system involves two interconnected *Regions*. Initially, we estimate the circuitry of the first *Region* by adhering to the methodology outlined in Experiment 4, as presented in Fig 8 of Section 4.4. Following this, we proceed with both local and global circuit estimations for the extended system, utilizing priors derived from the initial estimation step. B and C. Simulated signals (CaI, VSDI, and BOLD signals) from ground truth (dotted lines) and estimated parameters (solid lines) are displayed. D and E. Comparisons between the ground truth parameters ($\theta^{gt}$, x-axis) and the estimated parameters ($\theta^*$, y-axis) are presented for the iterative (D) and the one-step (E) estimation schemes. The simulation results with optimal model parameters obtained from the iterative and the one-step estimation schemes are compared in (B) ~ (E).

where, $\sigma^2_{i,prior}$ and $\sigma^2_{i,post}$ represent the variance of prior and posterior distribution of parameter $i$.

We compared the posterior shrinkage of effective connectivity between the iterative and one-step estimation methods for both intra- and inter-regional connectivity. The results showed a significant reduction in variance from prior to posterior estimates in the iterative estimation process, particularly for intra-regional connectivity, as indicated by higher $\rho_i$ values in the blue box plots compared to the red box plots in Fig 10. In contrast, the one-step estimation method showed relatively less reduction in posterior variance for both intra- and inter-regional connectivity. These findings suggest that the iterative method more effectively utilized information from the data.

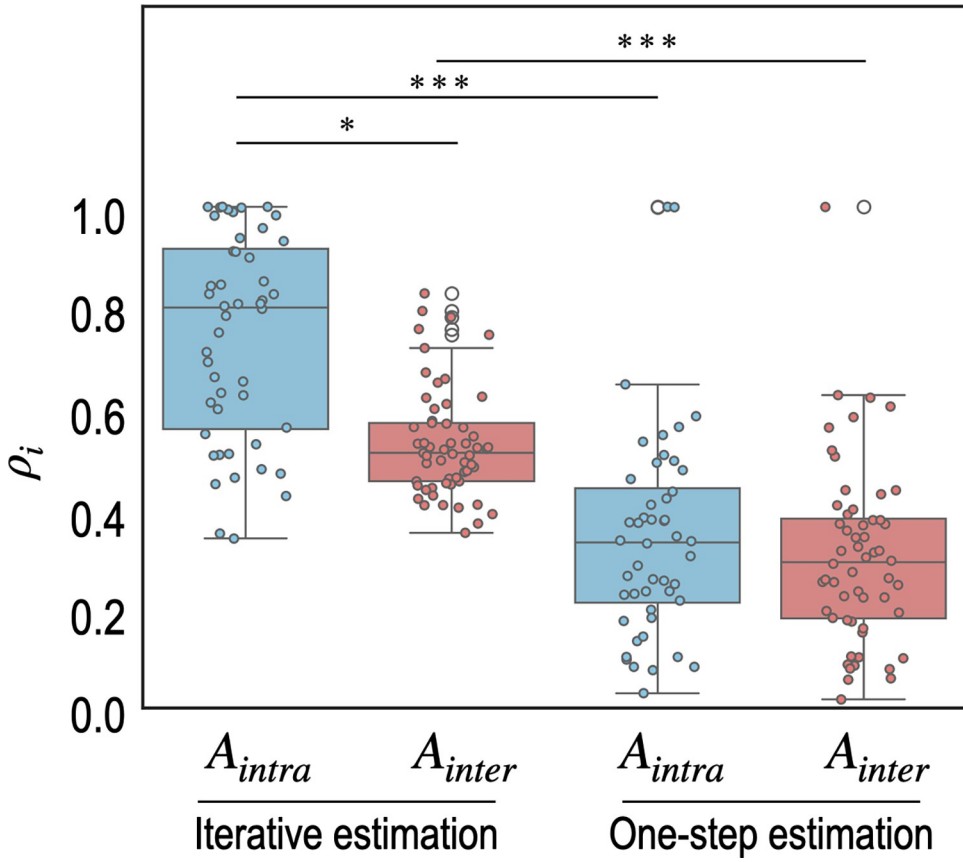

**Fig 10. Posterior shrinkages from prior distributions for each effective connectivity in experiment 5.** Posterior shrinkages ($\rho_i$) of each parameter $i$ from prior distributions in iterative estimation (left) and one-step estimation (right) are displayed. Both panels use blue box plots for intra-regional ($A_{intra}$) and red for inter-regional ($A_{inter}$) connectivity, with significant differences marked by three asterisks (* and *** represents $p < 0.01$ and $p < 0.0001$, respectively). Statistical significance was assessed using the Kruskal-Wallis test since the groups have different variances, violating the homogeneity of variance assumption required for one-way ANOVA.

## 5. Discussion

In this study, we present a DCM framework designed to effectively estimate neural circuits from multiscale and multimodal data, named mms-DCM. By utilizing a biologically plausible virtual system as a ground truth model, we illustrate how a reciprocal, multiscale approach that leverages diverse and partially observed multimodal signals can significantly enhance the accuracy of parameter estimation of a larger neural circuit. This approach strategically uses the posterior probabilities of local circuit parameters as informative priors for estimating global circuit parameters and, conversely, incorporates global circuit information to refine local circuit estimations. The framework is demonstrated using a combination of VSDI, CaI, and BOLD signals collected from a unified ground truth system. This innovative method offers a more nuanced and comprehensive understanding of neural circuits by effectively integrating varied neuroimaging data and may serve as a valuable tool for advancing our understanding of complex brain dynamics.

The increasing focus on multimodal signals in neuroscience research stems from their ability to provide complementary information crucial for understanding the brain. Studies like those by Engemann, Kozynets [32] and Schirner, Rothmeier [33] demonstrate the benefits of

combining modalities such as magnetoencephalography (MEG) and fMRI, or EEG and fMRI, for brain state modeling. These combinations are straightforward in terms of integrating the spatial coverage of EEG/MEG and fMRI data.

However, in preclinical research, the practical challenges of integrating multimodal data are more pronounced due to differences in resolution, span, and data availability across modalities. For instance, CaI typically focuses on a narrow region with cell-level resolution, whereas fMRI covers broader areas with regional-level resolution. This disparity in spatial coverage and resolution between modalities like CaI and fMRI presents a significant challenge. How to effectively combine such multimodal data with heterogeneous regional coverage is an issue that has not been extensively explored.

Addressing this challenge requires innovative computational approaches that can accommodate the differing scales and resolutions of various imaging techniques. Such models need to account for the unique strengths and limitations of each modality to provide a comprehensive and accurate representation of brain activity. Wei, Jafarian [10] proposed a DCM scheme using EEG-fMRI data, suggesting a Bayesian fusion method; a posterior density that was estimated from EEG was used as a prior of model parameters with fMRI data. In their approach, the neural state dynamics differ between EEG and fMRI. Unlike their approach, our study employs a common neural state dynamic model with different observation models, avoiding issues in matching parameters of different neural state models. We fit all available signals to estimate a single set of model parameters for the common neural state model.

We further propose an iterative methodology to combine neuroimaging data of varying scales and spatial extents, a common scenario in preclinical and basic neuroscience research. This method effectively addresses the typical trade-offs between high-resolution imaging, which focuses on specific areas, and low-resolution imaging that covers the entire system. Our approach, as exemplified by combinations like CaI with VSDI or CaI with BOLD, leverages parameter posteriors estimated at one scale as priors for another scale. For instance, local circuits estimated with CaI data in a specific region are used as priors for broader system estimation with VSDI, and similarly in reverse.

Our experimental results demonstrate that this iterative estimation approach is more effective than traditional one-step estimations that rely on less informative priors. This process allows for a more constrained and biologically plausible search space for the larger circuit model, enhancing the accuracy of the neural circuit estimation. Notably, this methodology underscores the importance of integrating both local and global circuit information from different imaging modalities. By recognizing the reciprocal and complementary roles of these diverse data sources, we can more accurately model neural activity, accommodating the heterogeneity inherent in the observation modalities.

Theoretically, there is no inherent prioritization between estimating local intra-regional and global inter-regional parameters; instead, iterative estimation facilitates convergence by allowing these parameter sets to inform each other reciprocally. Practically, however, it is preferable to first estimate low-level biological features, which then serve as a basis for establishing global parameters. We assumed that local microscopic biological properties are generally consistent across different brain regions compared to the more variable inter-regional global parameters. Based on this assumption, we assigned priors to parameters for regions without observed signals using parameter estimates from regions with observed signals, thereby enhancing the efficiency of the entire parameter estimation process. Additionally, under this assumption, we used BAYESopt to search for reference values for local properties. By using a single reference value for each biological property across all local regions, this approach constrains parameter estimates for these properties within a similar range across regions, even after final DCM fine-tuning. Conversely, suppose we do not use a single reference value but

instead use DCM with a wide prior variance for each biological property in each region. In that case, we cannot ensure that biological properties remain within a similar range across local regions.

In multiscale brain modeling, a common approach has been the bottom-up method, where small modules with limited parameters are compiled, often sharing parameters across small modules to fit brain dynamics data. For example, Schmidt, Bakker [34] developed the full-density multi-area spiking network model of the macaque visual cortex, successfully reproducing cortical activity features across scales, from individual cell spiking statistics to global resting-state networks. In their model, microcircuits for each cortical area, detailed in their layer-specific architecture and connectivity, were constructed from smaller modules, reflecting the intricate dynamics of brain activity on various scales. Likewise, Dura-Bernal, Suter [35] developed the NetPyNE tool (https://www.netpyne.org), which allows for the construction of detailed multiscale models by specifying high-level rules and parameters, which are then automatically translated into NEURON (https://nrn.readthedocs.io/en/8.2.4/) simulation components. This method simplifies the process of incorporating complex experimental data from various scales, from molecular to cellular to network levels, into unified computational models. This approach generally focused on the rule-based bottom-up model construction and relatively less emphasis on large circuit parameter optimization.

Our study, however, diverges from these bottom-up multiscale methods by employing an integrative strategy. We utilize both bottom-up and top-down approaches that effectively harness prior knowledge, leveraging the insights gained from available signals to inform the parameters of unobserved signals. The findings of our study reinforce the notion that local neural activities are deeply entangled with and significantly influence global circuit dynamics. This interplay highlights the importance of understanding global circuits for accurately estimating local circuits, as local regions do not operate in isolation. The accurate modeling of local circuit activities necessitates understanding their interactions within the broader neural network. Conversely, information about local circuits is equally crucial when estimating larger, global circuits. Data observed in local circuits can provide critical insights into the nodal properties of other regions within the global neural network, particularly in areas where direct observations might not be available. This approach leads to more biologically accurate estimations of parameters across the neural network. In our experimental framework, a local circuit, defined by experimental data from a specific region, serves as a model or reference for other regions, allowing for slight variations across local regions. This is crucial for capturing the subtle differences and nuances that exist in different parts of the neural network while maintaining a coherent model structure.

The primary limitation of our study is that its validity has been tested only in simulation-based experiments, which may not fully encapsulate the complexities of real-world scenarios. While our simulations are informed by data from the mouse cortical circuit, extending this methodology to experimental data involving multiscale and multimodal dimensions remains an objective for future research. This goal is contingent on the availability of such data. Even with the application to experimental data, simulations might still play a crucial role due to the inherent difficulty in accessing ground truth in real neural systems [10,36–38]. Additionally, the simplification that local neural circuits are uniform across different brain areas may introduce inaccuracies, especially within large and heterogeneous neural systems. Furthermore, given the higher number of parameters relative to observational data, particularly in the nonlinear system, there is a risk that different effective connectivity configurations could yield similar observational outcomes, introducing potential degeneracy, as discussed in prior studies [39,40]. Thus, the primary challenge is not reaching an optimal solution but ensuring that this solution is valid and accurately represents real biological systems. Addressing this would

require a more precise model with additional biological constraints. These limitations pertain more to the selection of an appropriate model than to the parameter optimization approach addressed in this study.

Our study predominantly utilizes CaI, VSDI, and BOLD signals, which are commonly used in animal research. However, the fundamental principles of our approach have the potential for adaptation to human brain studies. For instance, this method could be applied using combinations of electrocorticogram (ECoG) or stereo-electroencephalography (SEEG) with EEG or MEG signals for large-scale whole-brain circuit analysis, particularly in the study of epilepsy. For this purpose, fiber tractography using diffusion magnetic resonance imaging can be combined for better modeling of the whole brain circuit.

Furthermore, while our current framework is built on a convolution-based model, its versatility allows for extension to other types of neural state models. An example of this adaptability can be seen in the conductance model demonstrated by Shaw, Muthukumaraswamy [41]. Additionally, it can be adapted for a multi-state neural dynamics model, as explored in the context of diverse seizure states [42]. This flexibility suggests that our approach could be refined and expanded in future research to accommodate a broader range of neural dynamics and conditions.

In summary, our research presents the mms-DCM scheme, a novel approach in the construction of computational neural circuits for a large scope with details. We have shown that by integrating stepwise techniques and dynamically updating priors for both local and global circuit parameters driven by partially observed signals from multimodal imaging, our approach can effectively estimate the complexities of extensive neural systems. The strength of the mms-DCM lies in its ability to integrate diverse data sources and scales, overcoming mismatching of data availability, and providing a more holistic view of neural function and interactions.

## Supporting information

**S1 Text. Supporting Information.** The supplementary document explains the methodologies for extracting and analyzing calcium imaging signals from the barrel cortex, which were used to create a ground truth system.
(PDF)

## Author Contributions

**Conceptualization:** Jiyoung Kang, Hae-Jeong Park.

**Investigation:** Jiyoung Kang.

**Methodology:** Jiyoung Kang, Hae-Jeong Park.

**Supervision:** Hae-Jeong Park.

**Visualization:** Jiyoung Kang, Hae-Jeong Park.

**Writing – original draft:** Jiyoung Kang, Hae-Jeong Park.

**Writing – review & editing:** Jiyoung Kang, Hae-Jeong Park.

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
