## [Decision Letter · Decision Letter 0]

7 Oct 2024

Dear Dr Park,

Thank you very much for submitting your manuscript "Integration of Partially Observed Multimodal and Multiscale Neural Signals for Estimating a Neural Circuit Using Dynamic Causal Modeling" for consideration at PLOS Computational Biology.

As with all papers reviewed by the journal, your manuscript was reviewed by members of the editorial board and by several independent reviewers. In light of the reviews (below this email), we would like to invite the resubmission of a significantly-revised version that takes into account the reviewers' comments.

We cannot make any decision about publication until we have seen the revised manuscript and your response to the reviewers' comments. Your revised manuscript is also likely to be sent to reviewers for further evaluation.

Sincerely,

Daniele Marinazzo

Section Editor

PLOS Computational Biology

Daniele Marinazzo

Section Editor

PLOS Computational Biology

Reviewer's Responses to Questions

**Comments to the Authors:**

Reviewer #1: This study introduces a dynamic causal modeling (DCM) method to address the challenge of integrating unevenly observed, multiscale signals within a larger-scale neural circuit. This is achieved by employing a shared neural state model and observation models for different imaging modalities. The authors propose a method that reciprocally integrates local and global information from different modalities to infer the target circuit.

The proposed reciprocal and iterative technique substantially improves the estimation of local and global connectivity compared to traditional one-step estimation approaches, which typically rely on less informative priors for unobserved data.

This study raises important questions, particularly relevant to the DCM community, where causal inference is widely applied across different domains. The results are interesting, but the paper needs more clarity regarding what has been accomplished. Specifically, the introduction and abstract are written in a very generic manner, which raises issues related to what exactly has been done. From the introduction, I was left with the impression that one could infer calcium imaging (CaI) from fMRI data, which seems misleading!!

If local and global information informs each other, should there not be a sequential relationship between them, or can they be decoupled? Why are they not inferred simultaneously from the data?

In the introduction, the phrase "combining unevenly observed multimodal signals" is unclear—how does this apply? In terms of dimension? It wasn't clear from the introduction what is meant by multimodal multiscale inference. As a reader, should I expect to see the inference of CaI from VSDI or fMRI? Does this involve all the activity of CaI or only certain signals? Combining them could indeed improve model evidence, but the introduction gives the impression that one can infer CaI from EEG/fMRI or, more generally, infer fMRI from EEG/MEG, which seems overly ambitious. If this works, does it mean one can generate recordings that we currently don’t know how to measure?

My main concern is: if EEG and fMRI data are combined into a single system (shared neural state model), then the system must be dynamically rich and multiscale (i.e., involve slow and fast timescales, up/down states, and limit cycles). How is this feasible? Wouldn't you need different mappings (gain matrices) from source to sensors for each modality? Are these mappings known? How are the modalities coupled in the state space? For example, the Epileptor model offers a taxonomy of seizures (Jirsa et al., Brain 2014, 10.1093/brain/awu133), but this requires strong a priori knowledge about the combined modalities. It would be very helpful to formulate the problem in a state-space framework and clearly define what is known, what is observed, and what is inferred (with dimensions included).

For Figure 2, it would be helpful to show true versus estimated effective connectivity.

For Table 3, it would be beneficial to describe the estimations—if they are the same, perhaps in the caption, but if they differ, include a column for each experiment.

How strong is the prior, and what is the gain of information from prior to posterior? For instance, Hashemi et al. introduced a fully Bayesian method to examine the influence of priors on estimation (https://doi.org/10.1371/journal.pcbi.1009129).

What are the limitations of the study, and under what conditions does the algorithm fail (in-silico)? For example, in terms of parameter/variable dimensions? SEEG signals are sparse, and inference from them is a formidable challenge. The approach used in this paper could be very helpful in informing non-identifiable parameters.

Lastly, is there any investigation or comment on the degeneracy (or non-identifiability) of effective connectivity?

Reviewer #2: This study addressed an ambitious modelling challenge: how to infer the common neural dynamics that give rise to neuroimaging measurements at different spatial and temporal scales, specifically calcium imaging, voltage-sensitive dye imaging and BOLD fMRI. The novelty of the authors’ solution is 1) the pairing of a single neural model with modality-specific observation models, and 2) implementation of Bayesian model inversion methods, specifically the initialization of parameters using a scheme called Bayesian Optimization, and the design of an efficient scheme for iteratively estimating local and global parameters. This is an excellent piece of work and a strong contribution to the field. I just have a single query / suggestion for the authors, that relates to their novel scheme for model parameter estimation (3.1.2).

The authors began model estimation using Bayesian Optimization (BAYESopt) to select suitable priors for subsequent estimation using the typical Variational Laplace (VL) scheme in DCM. It took me a bit of time to understand this. I suggest adding a paragraph to the methods section introducing Bayesian Optimization and clarifying the distinction between this and the standard VL scheme (which of course is also a scheme for Bayesian optimization). In particular, could the authors explain whether BAYESopt offer some advantage over simply relaxing the prior variances to make them uninformative and using the VL scheme?

**Have the authors made all data and (if applicable) computational code underlying the findings in their manuscript fully available?**

Reviewer #1: Yes

Reviewer #2: Yes

PLOS authors have the option to publish the peer review history of their article (what does this mean?). If published, this will include your full peer review and any attached files.

Reviewer #1: No

Reviewer #2: No
---

## [Decision Letter · Decision Letter 1]

19 Nov 2024

Dear Dr Park,

We are pleased to inform you that your manuscript 'Integration of Partially Observed Multimodal and Multiscale Neural Signals for Estimating a Neural Circuit Using Dynamic Causal Modeling' has been provisionally accepted for publication in PLOS Computational Biology.

Best regards,

Daniele Marinazzo

Section Editor

PLOS Computational Biology

Daniele Marinazzo

Section Editor

PLOS Computational Biology

Feilim Mac Gabhann

Editor-in-Chief

PLOS Computational Biology

Jason Papin

Editor-in-Chief

PLOS Computational Biology

Reviewer's Responses to Questions

**Comments to the Authors: **

Reviewer #1: Thanks to author for the detailed responses. The revision deals with satisfactory results.

**Have the authors made all data and (if applicable) computational code underlying the findings in their manuscript fully available?**

Reviewer #1: Yes

PLOS authors have the option to publish the peer review history of their article (what does this mean?). If published, this will include your full peer review and any attached files.

Reviewer #1: **Yes: **MH

---

## [Editor Report · Acceptance letter]

4 Dec 2024

PCOMPBIOL-D-24-01465R1 

Integration of Partially Observed Multimodal and Multiscale Neural Signals for Estimating a Neural Circuit Using Dynamic Causal Modeling

Dear Dr Park,

I am pleased to inform you that your manuscript has been formally accepted for publication in PLOS Computational Biology. Your manuscript is now with our production department and you will be notified of the publication date in due course.

With kind regards,

Lilla Horvath
